# Recurrent mutations drive the rapid evolution of pesticide resistance in the two-spotted spider mite *Tetranychus urticae*

Li-Jun Cao[1†], Jin-Cui Chen[1†], Joshua A Thia[2†], Thomas L Schmidt[2], Richard Ffrench-Constant[3], Lin-Xi Zhang[1,4], Yu Yang[1,5], Meng-Chu Yuan[1,5], Jia-Yue Zhang[1,6], Xiao-Yang Zhang[1,5], Qiong Yang[2], Ya-Jun Gong[1], Hu Li[4], Xuexin Chen[7], Ary A Hoffmann[2*], Shu-Jun Wei[1*]

[1]Institute of Plant Protection, Beijing Academy of Agriculture and Forestry Sciences, Beijing, China; [2]Bio21 Institute, School of BioSciences, University of Melbourne, Parkville, Australia; [3]Biosciences, University of Exeter, Exeter, United Kingdom; [4]Department of Entomology, College of Plant Protection, China Agricultural University, Beijing, China; [5]College of Plant Science and Technology, Beijing University of Agriculture, Beijing, China; [6]Hebei North University, College of Agriculture and Forestry Technology, Zhangjiakou, China; [7]Institute of Insect Science, Zhejiang University, Hangzhou, China

*For correspondence:
ary@unimelb.edu.au (AAH);
shujun268@163.com (S-JW)

†These authors contributed
equally to this work

Competing interest: The authors
declare that no competing
interests exist.

Reviewing Editor: Detlef Weigel,
Max Planck Institute for Biology
Tübingen, Germany

## eLife Assessment

This study provides **important** insights into the evolution of pesticide resistance, demonstrating that resistance can arise rapidly and repeatedly, which complements prior work on parallel evolution across species. The combination of extensive temporal sampling in the field, experimental evolution, and genomics makes for **compelling** findings. The authors are to be commended for acknowledging the main limitations of their study in the Discussion. Framing the work in a broader context of resistance beyond arthropod pests would further increase the appeal of the study, which is of relevance for both agronomic practitioners and evolutionary biologists.

**Abstract** The genetic basis of pesticide resistance has been widely studied, but the exact nature of this evolutionary process in the field is often unclear, particularly when a limited number of populations is considered and when there is a lag between the evolutionary event and its investigation. We showed that an unprecedented number of recurrently evolved mutations in an arthropod pest, the two-spotted spider mite *Tetranychus urticae*, drive the rapid evolution of resistance to a recently commercialized acaricide, cyetpyrafen. We first observed high levels of resistance that appeared and became widespread within three years. Genome scans revealed genetic heterogeneity of resistance among populations and identified 15 target mutations, including six mutations on five amino acid residues of subunit *sdhB*, and nine mutations on three amino acid residues of subunit *sdhD* of the pesticide target succinate dehydrogenase, with as many as five substitutions on one residue. No mutations were present in 2,317 screened historical specimens, suggesting that mutations arose rapidly through de novo substitutions or from very rare segregating mutations. Identical mutations recurrently appeared in different genetic backgrounds, increasing the likelihood of resistance evolution. The high number of mutational options available for the evolution of target site resistance in this pest challenges resistance management practices.

**eLife digest** Chemical pesticides have been widely used for over a century to manage crop pests and disease vectors. However, pests can rapidly evolve resistance to these chemicals, enabling them to survive exposures that would otherwise be lethal.

Since the first documented case of pesticide resistance in 1914, instances of insecticide resistance have grown rapidly, posing serious threats to agriculture, human health and the environment. Despite a lengthy history of resistance and related research, most studies have been carried out only after resistance became widespread. Consequently, our understanding of how resistance develops under field conditions remains limited.

Cao et al. studied the two-spotted spider mite, *Tetranychus urticae*, to understand how pesticide resistance can emerge. *T. urticae* is considered the most resistant invertebrate pest worldwide due to its ability to develop resistance to a wide range of pesticides. The researchers investigated how resistance to the new pesticide cyetpyrafen emerges in field populations by comparing samples collected before and after cyetpyrafen use. To test the mites' susceptibility to pesticides, they conducted bioassays on 46 mite population samples collected from various regions in China between 2020 and 2024 and compared the results to previously reported data of earlier collections between 2017 and 2018 during the early release stage of cyetpyrafen.

The experiments revealed that high levels of resistance to cyetpyrafen emerged and spread across geographically distant populations within just three years of its introduction. Genomic analysis identified 15 distinct mutations in the target protein of cyetpyrafen, which consisted of an unprecedented number of amino acid changes associated with resistance to a single pesticide in a single species. Mutations arising after the release of cyetpyrafen were absent in samples collected before cyetpyrafen use, suggesting they arose only following the pesticide's introduction. Identical mutations appeared independently in multiple populations, indicating that resistance can evolve repeatedly and rapidly. Even a single mutation was sufficient to lead to high levels of resistance.

Cao et al. provide a rare longitudinal view of resistance development in field populations of an invertebrate pest by demonstrating that resistance in *T. urticae* can arise through multiple genetic pathways rather than solely through the spread of a single rare mutation. These findings underscore the challenges of managing pesticide resistance and highlight the urgent need for alternative pest control strategies and more considered pesticide use.

## Introduction

The development of pesticide resistance represents rapid contemporary evolution of organisms in response to human-induced stressors (*Ffrench-Constant et al., 2004*, *McKenzie and Batterham, 1994*; *Tabashnik et al., 2013*; *Gould et al., 2018*). Since the first instance of pesticide resistance in a scale insect was documented in 1914, cases have exponentially increased in arthropods, weeds, and pathogens, posing threats to agricultural production, human health, and the environment (*Sparks et al., 2021*; *Melander, 1914*; *Gould et al., 2018*; *Fritz, 2022*; *Peterson et al., 2018*). This has led to attempts to stop the evolution of resistance, or at least slow its appearance within populations (*Gould et al., 2018*). A key task in resistance management is to understand the evolution of resistance in field populations, where a given mutation may have originated once and spread via gene flow or evolved through multiple independent substitutions, and where resistance-associated mutations can pre-date (standing genetic variation) or arise subsequent to (de novo mutation) the introduction of pesticide selection pressures (*Ffrench-Constant, 2007*; *Daborn et al., 2002*; *Kersten et al., 2023*).

A challenge in understanding the evolution of resistance is that the genetic basis of pesticide resistance observed in laboratory-generated resistant strains often does not reflect that in field populations, especially for arthropod pests (*Chen et al., 2023*; *Walsh et al., 2022*; *ffrench-Constant, 2013*; *Legan et al., 2024*). Field populations are typically much larger and more genetically heterogeneous than laboratory populations. Because of the greater potential availability of rare mutations with large effect sizes in field populations, monogenic resistance to pesticides involving these rare mutations may occur relatively more frequently than in laboratory-generated strains (*McKenzie and Batterham, 1994*), though small effect loci can also remain important in field resistance (*Schmidt et al., 2024*; *Lucas et al., 2023*). In contrast, artificial selection in the laboratory usually takes place in populations

with much smaller effective population sizes, as well as a constant intensity and pattern of selection expected to favor the accumulation of multiple changes of small effect (*ffrench-Constant, 2013*). Where alleles with large effects are selected in the laboratory, these may not necessarily reflect the full set of resistant mutations in the field (*Legan et al., 2024*). Such mismatches between laboratory-selected and field-evolved resistance may lead to an inaccurate understanding of resistance evolution (*ffrench-Constant, 2013*; *Bras et al., 2022*; *Legan et al., 2024*).

A further challenge in understanding resistance evolution in the field is that the genetic basis of resistance is often examined retrospectively well after the emergence of resistance in the field. Retrospective studies can fail to capture processes that have occurred in the intervening period (*Andreev et al., 1999*; *Raymond et al., 1991*; *ffrench-Constant et al., 1993a*; *Adesanya et al., 2021*), such as the introduction of new alleles through gene flow or de novo mutations, and fitness costs as well as changing selection regimes through pesticide rotations that can lead to resistance alleles being transient (*Schmidt et al., 2010*; *ffrench-Constant, 2013*; *Kreiner et al., 2022b*; *Kreiner et al., 2019*; *Groeters and Tabashnik, 2000*). Preserved specimens before and following the introduction of pesticides can allow for resistance processes to be investigated retrospectively, but samples are often unavailable (*Hartley et al., 2006*; *D'Costa et al., 2011*). Ongoing genetic monitoring of populations can help in overcoming some of these issues, particularly if population processes affecting a species are understood and if the genetic basis and new resistant alleles are identified as they evolve in the field (*Van Leeuwen et al., 2012*; *Pélissié et al., 2022*; *Pezzini et al., 2024*; *Taylor et al., 2021*).

Here, we take advantage of the recent evolution of resistance in the two-spotted spider mite (TSSM), *Tetranychus urticae* (Acari: Tetranychidae), to a novel acaricide, cyetpyrafen, to provide a detailed understanding of contemporary resistance evolution. TSSM is a global super pest that has evolved resistance to at least 11 groups of acaricides based on their modes of action and is the most resistant invertebrate pest worldwide (*De Rouck et al., 2023*; *Van Leeuwen et al., 2010*). Succinate dehydrogenase inhibitor (SDHi) acaricides (IRAC group 25) were first launched in 2007 for mite control. Four chemicals consisting of cyetpyrafen (first launched in 2017 in China), cyenopyrafen (2009 in Japan), cyflumetofen (2007 in Japan), and pyflubumide (2015 in Japan) fall within this group of acaricides globally (*Hayashi et al., 2013*; *Nakahira, 2011*). Laboratory-selected resistance has already been identified to these new acaricides and is conferred by target-site mutations of the SDH coding *sdhB* and *sdhD* genes, with mutations identified at three positions (*Njiru et al., 2022*; *Sugimoto et al., 2020*). In China, three acaricidal SDH inhibitors have been registered over the last decade: cyflumetofen (2013), cyetpyrafen (2017), and cyenopyrafen (2019). Since its release in 2017, cyetpyrafen has become the most popular pesticide for controlling spider mites, though low-level resistance was detected in the field only one year after its release (*Chen et al., 2019*).

We have now tracked resistance through repeatedly monitoring field-evolved resistance of TSSM to cyetpyrafen at a time when resistance has developed rapidly to high levels across geographically distant populations. We investigated resistance using a combination of population genomics, laboratory selection, bioassays, and studies on historically preserved specimens before the introduction of cyetpyrafen. We find that multiple and recurrently evolved mutations drive the rapid development of high-level resistance to cyenopyrafen in field populations across a widespread geographical range. We also demonstrate that these mutations likely evolved de novo after the introduction of cyetpyrafen, though they may also have originated from extremely rare mutations segregating in populations before cyetpyrafen selection. We emphasize the benefits of integrating genomics with other approaches in longitudinal studies to capture the dynamics of rapid evolutionary changes.

## Results
### Strong resistance evolves rapidly in field and lab-selected populations

To monitor the evolution and spread of cyetpyrafen resistance in the field, we conducted bioassays on 46 mite population samples collected from various regions in China between 2020 and 2024 (*Figure 1*, *Supplementary files 1 and 2*). The susceptibility of these population samples to cyetpyrafen was compared with previously reported data for populations collected between 2017 and 2018 during the early release stage of cyetpyrafen in China (*Chen et al., 2019*; *Gong et al., 2017*). In 2017–2018, all seven tested populations remained susceptible, with $LC_{50}$ values ranging from 1.4 to 21.6 mg/L (*Figure 1a*). In 2020, one population from Shouguang City, Shandong Province (SDSG2),

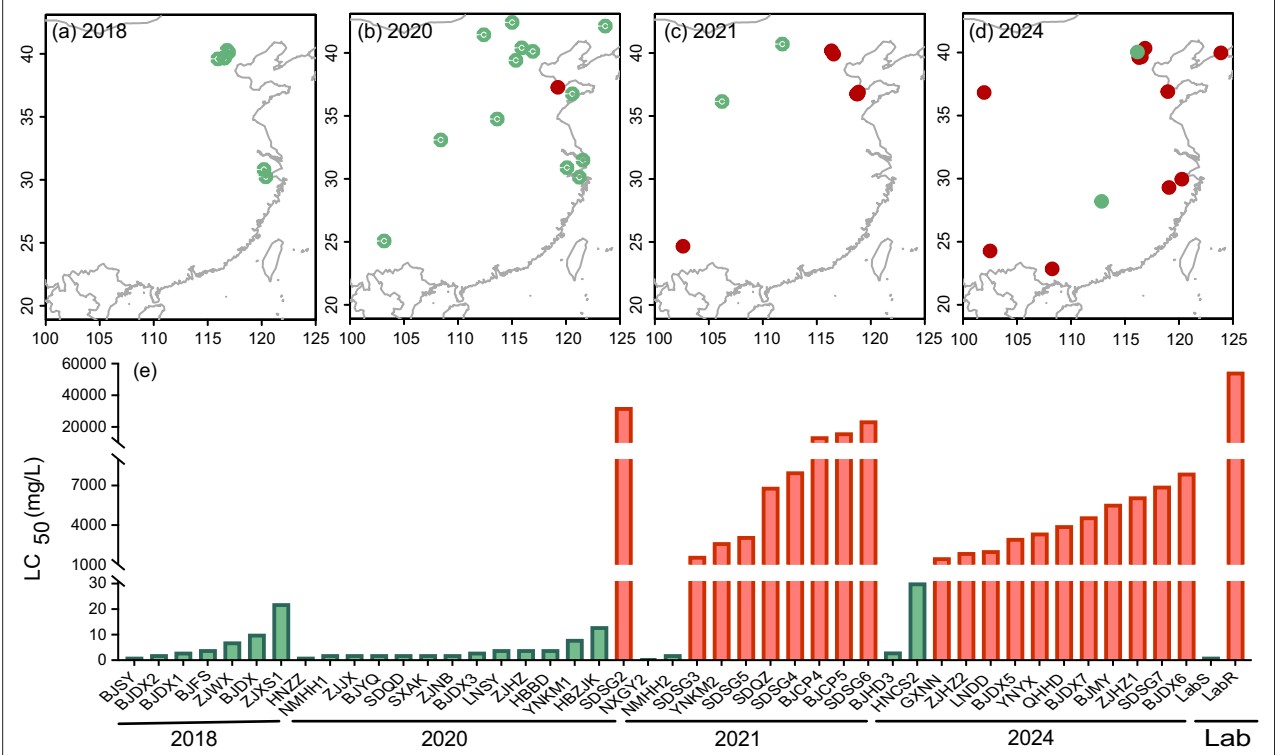

**Figure 1.** Susceptibility of the two-spotted spider mite *Tetranychus urticae* to cyetpyrafen. (**a–d**) Collection maps of populations used for bioassay in different years. Red and green points represent resistant and susceptible populations, respectively. (**e**) $LC_{50}$ values of each population collected in 2018 (7 populations) (*Chen et al., 2019*), 2020 (14), 2021 (10), 2024 (13), and a laboratory-selected resistant population (LabR) and a parallel reared susceptible population (LabS) to cyetpyrafen. The green bars show the susceptible populations, while the red bars show the resistant populations.

showed resistance with an $LC_{50}$ >32,000 mg/L, while the other 13 populations remained susceptible (*Figure 1b*). By 2021, high levels of resistance to cyetpyrafen quickly became widespread in field populations: eight of 10 tested populations had an $LC_{50}$ higher than 1606.2 mg/L (*Figure 1c*). In 2024, 11 of the 13 tested populations exhibited resistance, with an $LC_{50}$ higher than 1498.6 mg/L (*Figure 1d*). Overall, the susceptibility of all populations was bimodal, with populations being highly susceptible ($LC_{50}$ ranges from 0.61 to 30.1 mg/L, $\bar{x}$=6.72 mg/L) or highly resistant ($LC_{50}$ >1498.6 mg/L, $\bar{x}$>5892.5 mg/L) and an 877-fold difference between mean $LC_{50}$s of the susceptible and resistant groups. While only a single sample was resistant in 2020, by 2021 resistance was observed across the entire range of China (*Figure 1c*). This pattern indicates that field populations of TSSMs have rapidly developed high levels of resistance to cyetpyrafen across a wide geographical area.

We conducted artificial selection to further investigate resistance evolution, using two laboratory strains derived from the same field population and reared in parallel. The field population for selection was collected from Hangzhou, Zhejiang province (ZJXS1), with an $LC_{50}$ of 21.6 mg/L. One laboratory strain, 'LabR', was exposed repeatedly to cyetpyrafen, while the other, 'LabS', was not. LabR underwent a gradual increase in cyetpyrafen resistance for the first 15 selection rounds (31 generations), but a much sharper increase over the remaining 18 selection rounds (35 generations) (*Supplementary file 3*). By the end of the experiment, LabR possessed an $LC_{50}$ of 54,336 mg/L, a 2521-fold increase in resistance relative to the ZJXS1 field population. In contrast, the $LC_{50}$ of the LabS strain decreased to 1.21 mg/L over the same period (*Figure 1e*, *Supplementary file 3*), due to unknown reasons, perhaps linked to adaptation to the laboratory environment. However, this is a very minor change in resistance compared to the large increase seen in the selected strain.

Additional bioassays were used to test whether the selected LabR strain showed cross-resistance to other pesticides that target proteins in the mitochondrial electron transport chain. For the other two mitochondrial complex II inhibitors (IRAC group 25), the $LC_{50}$ of cyenopyrafen increased 21,254-fold (3.76–79,918 mg/L), and the $LC_{50}$ of cyflumetofen increased 69,861-fold (6.47–452,003 mg/L).

For the mitochondrial complex III inhibitors, the $LC_{50}$ of bifenazate (group 20D) increased twofold (9.87–18 mg/L), and the $LC_{50}$ of pyridaben (group 21 A) increased threefold (3,763.4–10,352 mg/L). The results indicate that cyetpyrafen selection led to high cross-resistance to other mitochondrial complex II inhibitors but weak cross-resistance to the mitochondrial complex III inhibitors (*Supplementary file 3*).

## Selective sweep signals are clear but heterogeneous

To identify genes potentially associated with cyetpyrafen resistance, we conducted pooled whole-genome resequencing on 22 TSSM population samples, including historic samples of eight geographical populations collected in 2017 before the first field releases of cyetpyrafen, 12 samples (10 resistant and 2 susceptible) collected afterwards, and two laboratory populations (LabS and LabR) which originated from the same field population (*Supplementary files 1 and 4*). To allow for variation in the genetic basis of resistance and genetic backgrounds among populations, we chose the Population Branch Excess (PBE) analysis (*Yassin et al., 2016*) method to scan selection signals across the genome; we used two susceptible populations (NMHS2 and LabS) as non-focal populations to correct for the varied genetic backgrounds, and one resistant population as the focal population to identify population-specific signals of pesticide selection. No clear and coincident selection signal was identified among historical population samples (*Figure 2a*). The laboratory-selected resistant LabR population had clear selection signals at two genes, *sdhB* and *sdhD,* that encode two subunits of succinate dehydrogenase, the target of SDHi acaricides (*Figure 2b*). None of the 10 field-collected resistant populations had selection signals at both of these genes, but four (BJDX6, ZJHZ1, ZJHZ2, and LNDD) had selection signals at *sdhB*, and one (BJCP4) had a selection signal at *sdhD*. Five of the resistant populations (QHHD, GXNN, SDSG3, SDQZ, and YNYX) showed no selection signal at either gene; this same pattern was observed in the one population sample (HNCS2) that remained susceptible after cyetpyrafen was introduced. Additional PBE peaks were observed at other genomic regions in historical, resistant, and susceptible populations (*Figure 2a and b*); these could reflect demographic processes or other selective forces not considered in this study.

Selection for cyetpyrafen resistance could impact patterns of genetic diversity in the regions of interest or more widely across the genome if selection decreases overall population size. We investigated these impacts focusing on genome-wide patterns of nucleotide diversity ($\pi$) and on Tajima's *D*. Five resistant populations with selection signals identified by PBE analysis (*Figure 2b*) showed corresponding decreases in $\pi$ and Tajima's *D* near the two SDH genes (*Figure 2—figure supplement 1*). The overall genetic diversity and Tajima's *D* across the genome were not significantly associated with resistance level, genetic structure, or geographic coordinates (p>0.05; *Figure 2—figure supplement 2*, *Supplementary file 4*), suggesting that resistant populations had not experienced strong bottlenecks. Five resistant populations without selection signals did not show obvious reduced genetic diversity or altered Tajima's *D* values around the SDH genes (*Figure 2—figure supplement 1*), suggesting that these had not experienced strong selection or selective sweeps.

## Multiple mutations have arisen in the field

To identify specific mutations conferring resistance, we examined the two SDH genes, *sdhB* and *sdhD*, where selective sweep signals were detected in some populations. We used whole-genome resequencing data from pooled individuals; we also conducted Sanger sequencing of partial *sdhB* (216 bp of 1001 bp) and *sdhD* (155 bp of 1008 bp) genes to validate the pool-seq data and examined mutation frequencies in additional populations; this involved 1268 individuals from 52 populations, including samples collected before and after the release of cyetpyrafen (*Supplementary file 1*). Allele frequencies identified by pool-seq and Sanger sequencing were correlated (*Figure 3—figure supplement 1*). A total of 15 amino acid substitutions were identified, including six mutations on five amino acid residues of *sdhB*: H146Q (CAT to CAA), S212I (AGC to ATC), H258Y (CAT to TAT), I260T (ATC to ACC), I260V (ATC to GTC), and A285S (GCA to TCA; *Figure 3a*, *Supplementary file 2*), and nine mutations on three amino acid residues of *sdhD*: D116G (GAC to GGC), D116E (GAC/T to GAA), D116N (GAC to AAC), R119C (CGT to TGT), R119L (CGT to CTT), R119G (CGT to GGT), R119P (CGT to CCT), R119H (CGT to CAT), and P120L (CCC to CTC; *Figure 3b*, *Supplementary file 2*). The residue R119 on *sdhD* was a notable extreme case, involving five amino acid changes. The second codon position of R119 (G) was substituted by the other three nucleotides, while the first codon position (C) was

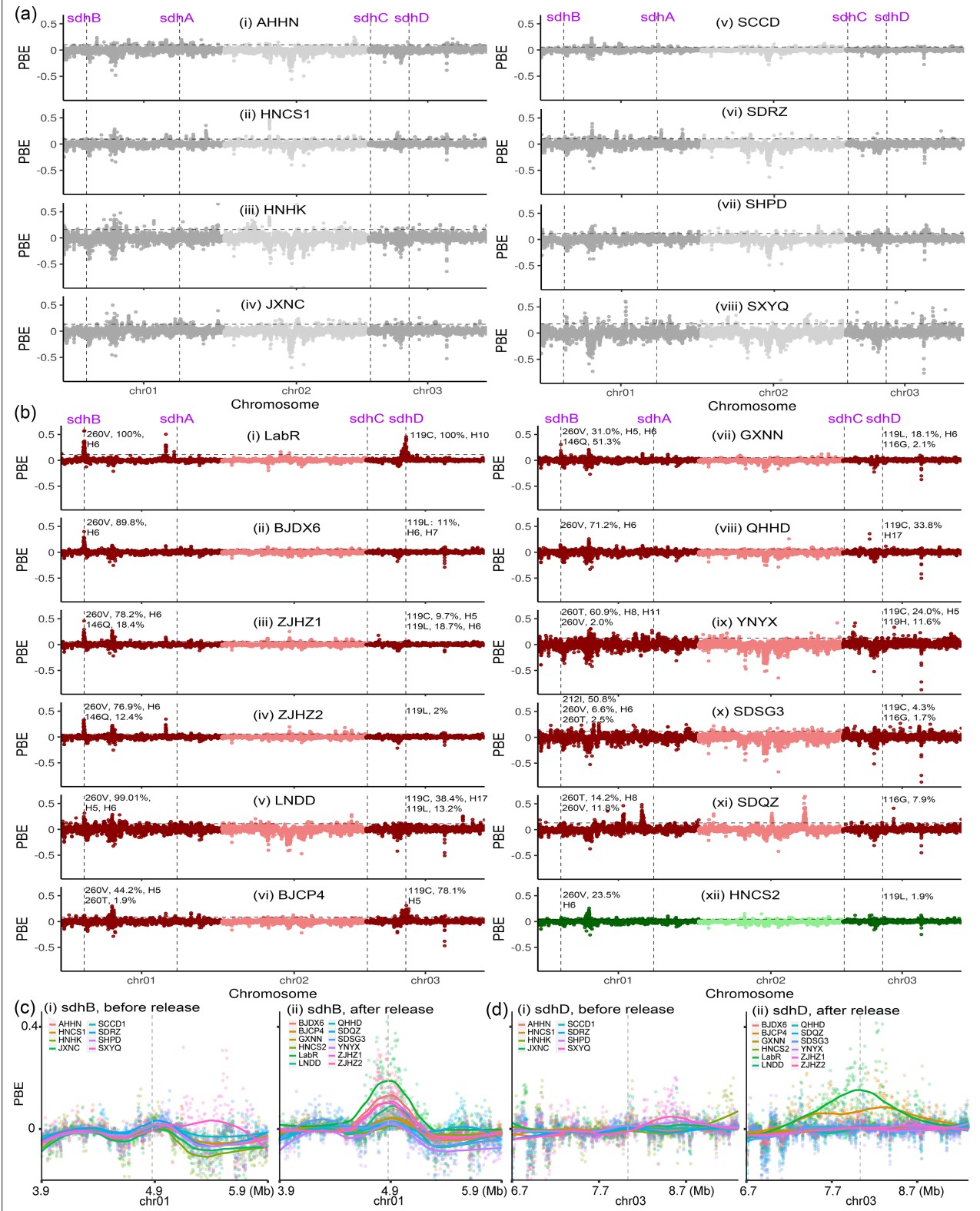

**Figure 2.** Signals of selective sweep across genomes of the two-spotted spider mite *Tetranychus urticae*. (**a**) PBE values across the genome for eight historic populations collected in 2017 before the commercial release of cyetpyrafen. No clear and consistent selection signal was identified among these samples. Chromosomes are colored alternately dark gray and light gray. (**b**) PBE values for one lab-selected resistant population (LabR), 10 resistant field populations, and one susceptible population (HNCS) collected in 2021 and 2024 after the release of cyetpyrafen. The laboratory-selected

*Figure 2 continued on next page*

*Figure 2 continued*

resistant LabR population had clear selection signals at two genes, *sdhB* and *sdhD*. Four (BJDX6, ZJHZ1, ZJHZ2, and LNDD) field-collected resistant populations had selection signals at *sdhB*, and one (BJCP4) had a selection signal at *sdhD*. Chromosomes are colored alternately dark red and light red for resistant populations, dark green and light green for susceptible population HNCS. Allele frequency (pool-seq data) and haplotypes of mutations are indicated near the corresponding *sdhB* and *sdhD* genes of each population in (**a**) and (**b**). (**c**) PBE values for genomic regions around the *sdhB* gene, comparing values between the historic (left) and recently collected (right) populations. (**d**) As above but for the *sdhD* gene. Each point represents a genomic window of 5-kp wide when compared to reference genomes involving the susceptible laboratory line (LabS) and a susceptible field population (NMHH2). The vertical lines indicate the position of *sdh* genes, while the horizontal lines represent the 1% threshold of PBE values.

The online version of this article includes the following figure supplement(s) for figure 2:

**Figure supplement 1.** Moving average of delta nucleotide diversity (**a - b**) and delta Tajima's *D* values (**c - d**) along chromosomes.

**Figure supplement 2.** Scatter plot of average genome-wide nucleotide diversity (π) and Tajima's *D* per population.

substituted by two other nucleotides (T and G), indicating that the first two codon positions of R119 are almost saturated in terms of possible nucleotide substitutions of these codon positions.

Multiple mutations were commonly found in a single population, while none of the mutations were fixed in any field population (*Figure 2b*, *Supplementary file 2*). Populations sampled in 2020 had at most two mutations, while those sampled in 2021 and 2024 had at most five. Populations sampled in 2020, 2021, and 2024 had, on average 1.17, 2.46, and 2.55 mutations, respectively. We also examined the other two SDH subunit genes, *sdhA* and *sdhC*, using pool-seq data, but no amino acid replacements were identified.

We checked the evolutionary conservation of these mutations across animals by aligning homologous sequences from ~700 animal species, encompassing nematodes, insects, fishes, and mammals (*Figure 3a and b*). The protein sequences of *sdhB* and *sdhD* were highly conserved among various animal species, particularly around the mutant residues H146, S212, I260, D116, and R119 observed in this study. In a structural model of human SDH, eight amino acid residues were predicted at the ubiquinone binding site (Q-site), corresponding to 211, 215, and 260 on mite subunit *sdhB*, 58, 63, 67, and 71 on *sdhC*, and 117 on *sdhD* (*Njiru et al., 2022*; *Du et al., 2023*). All detected mutations in the mite are located around or in the Q-site, although differences exist between human and mite sequences (*Figure 3c and d*).

Although cyetpyrafen only started to be used in 2017, it is possible that these potential resistance mutations were present as standing genetic variation before this time. To investigate this, we obtained Sanger sequencing and pool-seq data from 2,317 female mites collected before the cyetpyrafen release and 4631 after its release (*Figure 3e*, *Supplementary file 2*). Of the 15 putative resistance mutations, only one (A285S) was observed in the pre-release specimens, which was present in population SDRZ (unclear resistance status) but was not detected in specimens collected after the cyetpyrafen release. The historical trajectory of resistance alleles revealed extreme selection after release of cyetpyrafen (*Figure 3e*). These 14 resistance alleles have collectively experienced a selective strength of $\tilde{s} = 0.707$ (Z=3.372, p=0.0007) per year. However, based on logistic regression, the estimate of the strength of selection on each resistance allele was only significant for I260V (p=0.011). Seven mutations were first detected in 2020 (I260T, D116G, D116N, R119C, R119L, R119P, and R119H), three mutations were first detected in 2021 (H146Q, S212I, and I260V), and the remaining four mutations were first detected in 2024 (H258Y, D116E, R119G, and P120L; *Figure 3e*, *Supplementary file 2*). Among the target-site mutations, I260V and R119C were the most common, found in 65.9% and 46.3% (respectively) of the populations collected after the release of cyetpyrafen, with their frequency increasing rapidly (*Figure 3e*). In the laboratory-selected population LabR, two mutations (I260V and R119C) were fixed, but neither was detected in either the field population from which it was sourced (ZJXS1) or the unselected susceptible laboratory population derived from ZJXS1 (LabS). These results indicate that the putative resistance mutations were not likely to be present as standing genetic variation unless they were segregating at very low frequencies, and they may have evolved recently through de novo substitutions, although this cannot be demonstrated definitively.

## Recurrent evolution of identical mutations

When a specific resistance mutation is found in multiple populations, this can be the result of migration between populations or to multiple independent origins of the same mutation. We evaluated

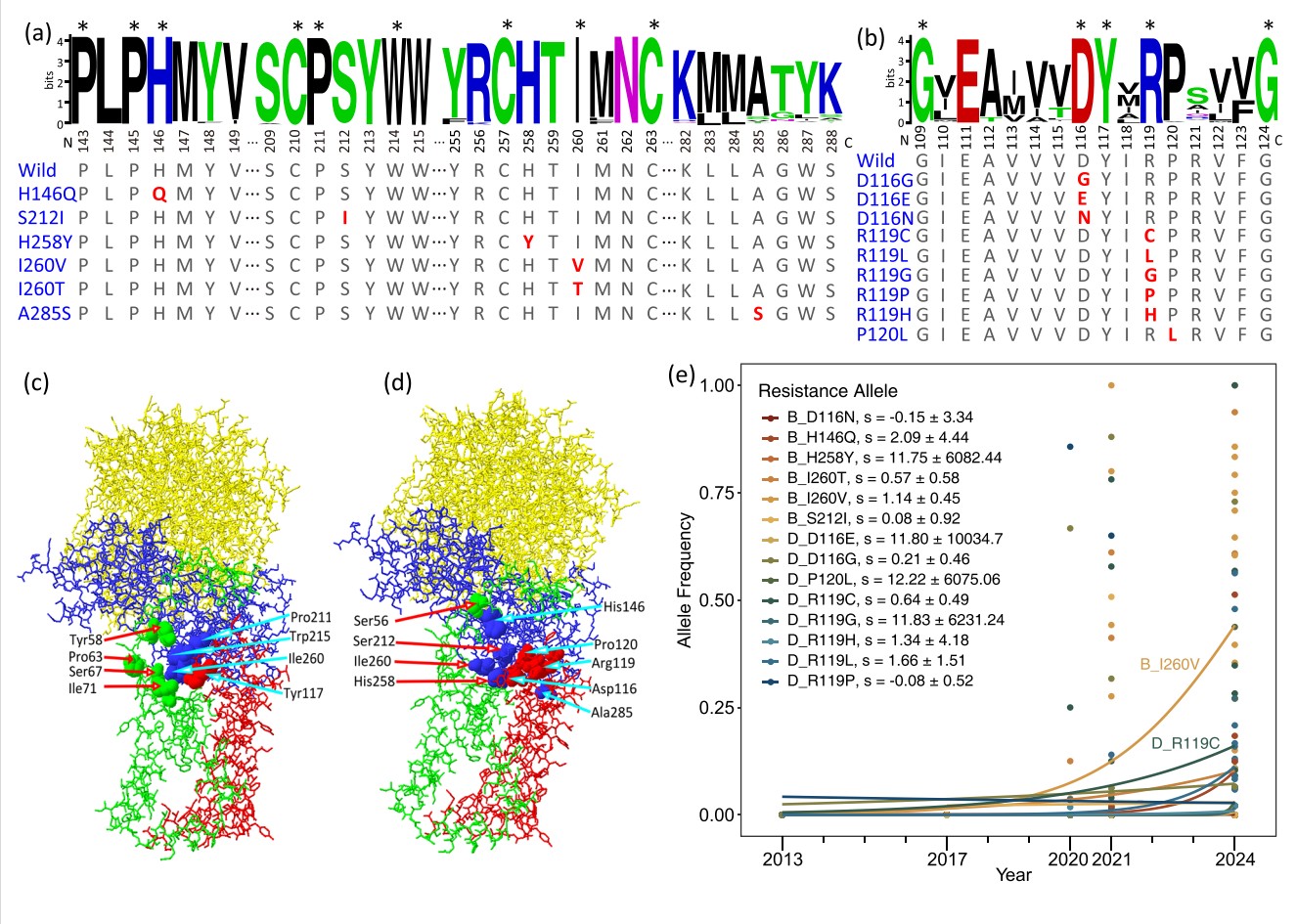

**Figure 3.** Amino acid mutations on *sdhB* and *sdhD* genes in populations of the two-spotted spider mite *Tetranychus urticae*. (**a, b**) Sequence logos at mutations related to cyetpyrafen resistance in *sdhB* (a, 757 sequences) and *sdhD* (b, 158 sequences). The numbers under logos indicate the position of amino acid residues on protein chains. Excluding *T. urticae*, conserved amino acids without any alternative are labeled with asterisks. Amino acids of *T. urticae* are shown below. Amino acids in red indicate mutations. (**c**) 3D structure of the wild-type succinate dehydrogenase of *T. urticae*. Solid 3D shapes show eight amino acid residues at the Q-site, which is the binding site of cyetpyrafen. The sdhA, sdhB, sdhC, and sdhD chains are shown in yellow, blue, green, and red, respectively. (**d**) Amino acid mutation sites on a 3D structure of the wild-type succinate dehydrogenase of *T. urticae*. Solid 3D shapes indicate mutations on five residues identified in this study (His258, Ile260, Asp116, Arg119, and Pro120) and one residue (Ser56) reported by previous studies (*Njiru et al., 2022*; *Sugimoto et al., 2020*) that confer resistance to SDH inhibitors. (**e**) The trajectory of allele frequencies of resistance mutations through time fitted by logistic regression. Each dot represents one population. A258S was omitted as it was only present in one population (SDRZ) sampled in 2017.

The online version of this article includes the following figure supplement(s) for figure 3:

**Figure supplement 1.** Correlation between allele frequencies genotyped by Sanger sequencing and pool-seq.

these evolutionary hypotheses using two methods: TreeMix (*Pickrell and Pritchard, 2012*) and haplotype network analysis.

We used TreeMix to investigate whether populations with identical alleles were more genetically similar or had recent histories of migration. TreeMix builds maximum likelihood trees from population allele frequencies and then adds optional migration edges between populations. We set the SCCD population from southwest China as the root population as it had the highest genetic diversity, and we built trees that followed a strict drift model (i.e. no migration) and trees with up to 10 migration edges. The strict drift model explained most of the variance in relatedness among populations (mean over 1000 replicates of 0.930). The best estimate of the number of historical migration events was one. The inclusion of a single gene flow event resulted in only a very small increase in explained genetic variance compared to 0.930 for the drift model, suggesting that migration among populations is low. The consensus tree over the 1000 replicates assigned most of the populations into northern and southern

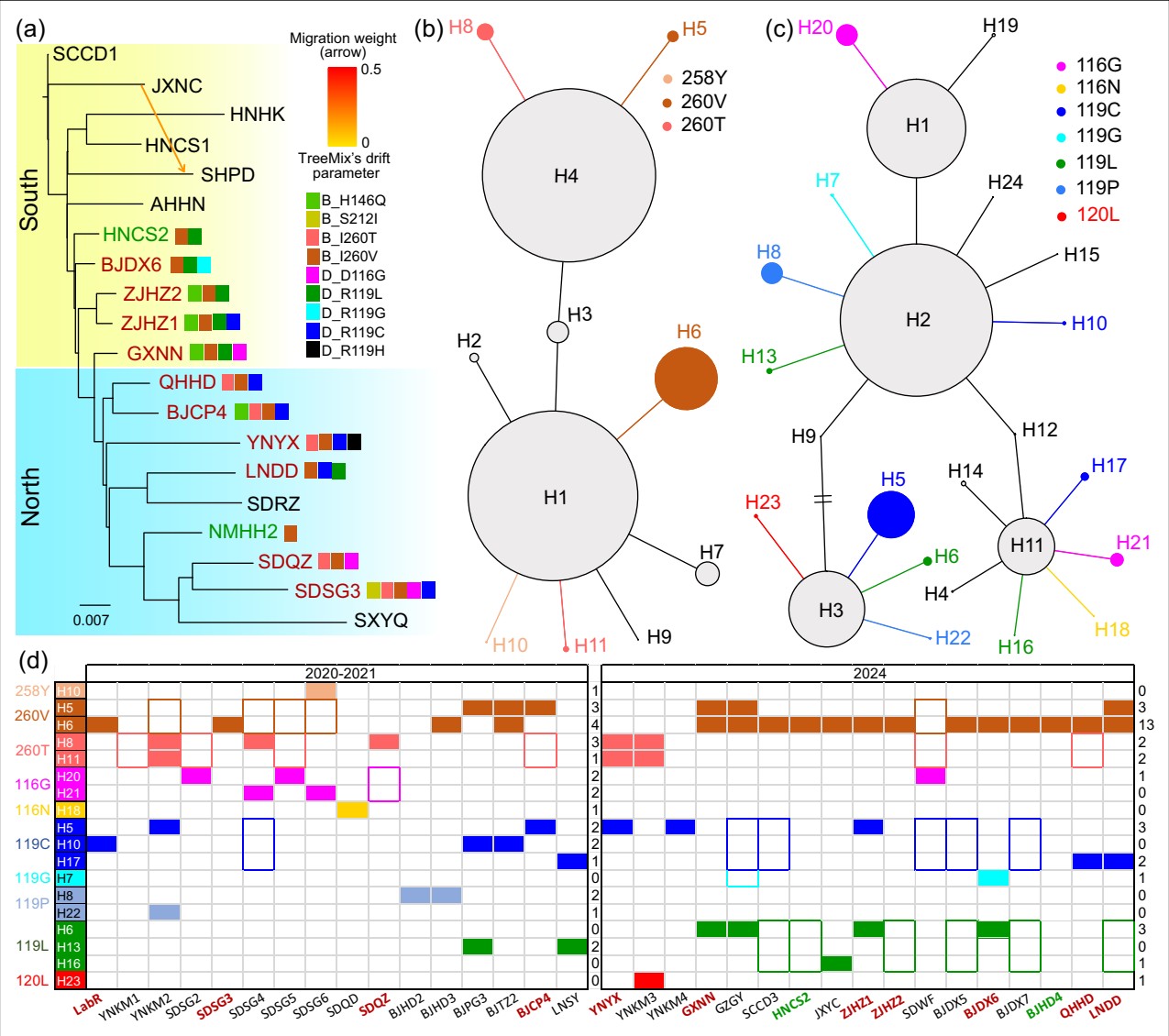

**Figure 4.** Genetic structure and haplotype networks of *sdhB* and *sdhD* genes of the two-spotted spider mite *Tetranychus urticae*. (**a**) Treemix result showing genetic structure and gene flow among populations. The mutations detected by pool-seq data were labeled near the corresponding population. (**b**) Haplotype network of *sdhB* gene inferred from homozygous individuals. In figures (**a**) and (**d**), the red population codes show resistant populations, the green ones show susceptible populations, while the black ones show populations of unknown resistance status. (**c**) Haplotype network of *sdhD* gene inferred from homozygous individuals. The colored haplotypes are those carrying the resistant mutation. (**d**) Presence of each haplotype carrying the target mutations in two periods. In the early resistance development stage of 2020–2021, each haplotype was mainly distributed across a limited range, while in 2024, the same haplotype was distributed across a wide geographical range. The filled rectangle indicates that the entire haplotype is present in the population, while a colored border suggests that only the mutations were detected, but the haplotype was not assigned. The number of populations carrying the corresponding mutation is listed in the right column for the two time periods.

The online version of this article includes the following figure supplement(s) for figure 4:

**Figure supplement 1.** Haplotypes of *sdhB* (**a**) and *sdhD* (**b**) genes.

groups by geography, but not by resistance status (*Figure 4a*). We also observed several instances where geographically distant populations grouped together. Some southern populations were placed with the northern group (YNYX), and vice versa (BJDX6); these populations may have been recently established via long-distance dispersal, possibly through the transport of seedlings.

The above-mentioned mutations in *sdhB* and *sdhD* genes were detected in six populations from the southern group and seven populations from the northern group. No significant isolation-by-distance pattern was detected in resistance allele frequencies across survey years (Mantel test: 2020, p=0.73;

2021, p=0.52; 2023, p=0.16), indicating that the spread of resistant alleles was not linked to geographical factors. The mutation I260V was detected in every field population sampled after the release of cyetpyrafen (*Figure 4a*, *Supplementary file 2*), based on pool-seq data. The mutation I260T was only detected in the northern group, and R119L was mostly detected in the southern group. The mutation H146Q was found in four adjacent populations (ZJHZ1, ZJHZ2, GXNN, and BJCP4). This geographical heterogeneity in the distribution of resistance mutations suggests multiple mechanisms of resistance evolution underlie these patterns, involving some spread via migration between populations (e.g. H146Q) as well as possible independent origins of the same mutation (e.g. I260V).

We explored these ideas further using haplotype network analysis. When resistance spreads via migration, these mutations should have identical haplotypes, unless recombination has taken place, which will introduce a second haplotype that segregates locally alongside the initial haplotype. We constructed the haplotype network for the partial *sdhB* (216 of 1001 bp; *Figure 4b*) and *sdhD* (155 of 1008 bp; *Figure 4c*) genes determined from Sanger sequencing. Since haplotypes could not be determined when two loci were heterozygous, we detected haplotypes from sequencing data with at most one heterozygous locus. In total, 844 and 696 individuals were used to detect haplotypes of *sdhB* and *sdhD*, respectively. The genes showed high genetic diversity, with 8 SNPs (*sdhB*) and 11 SNPs (*sdhD*) detected, which generated 11 (*sdhB*) and 24 (*sdhD*) haplotypes, respectively. We assumed that common haplotypes located at the center of the network are the ancestral haplotypes (H1 and H4 in the case of *sdhB*; H1, H2, H3, and H11 in the case of *sdhD*). We assumed all other haplotypes were derived. Each derived haplotype was only one nucleotide different from its nearest ancestral haplotype. Five of the nine derived *sdhB* haplotypes carried a putative resistance mutation, as did 13 of the 22 derived *sdhD* haplotypes. Combinations of these mutations between the two subunits were present. However, there was no instance of two or more resistance mutations occurring within the same haplotype.

Of the 10 putative resistance mutations detected through Sanger sequencing, all but four (H258Y, D116N, R119G, and P120L) were found as multiple haplotypes (*Figure 4b and c*). To test whether these haplotypes were produced by multiple origins or by recombination, we inferred the possible means by which recombination could produce each haplotype and investigated the geographical distributions of the requisite ancestral and derived haplotypes. For mutations in *sdhB*, we inferred that recombination would have to involve the H4 ancestral haplotype as donor, while the H11 ancestral haplotype was required for *sdhD* (*Figure 4—figure supplement 1*). Thus, recombination could have produced the *sdhB* haplotypes H5 (H4 x H6) and H8 (H4 x H11), and the *sdhD* haplotypes H17 (H11 x H10), H16 (H11 x H13), and H21 (H11 x H20). However, only three populations (GXNN, YNKM3, and YNYX) had both of the two recombining haplotypes required and one derived haplotype (*Supplementary file 5*). Additionally, the proximity of the mutations was such that multiple recombination events would be required within the same 5–60 bp regions, which seems unlikely even with a high recombination rate of (say) 10 cM/Mb where two mutations might only be expected to occur once in $2.5 \times 10^8$ effective individuals across 60 bp; this assumes a population recombination rate $\rho = 2 c_N e r$, where $r$ refers to recombination rate per base-pair per generation, $N_e$ refers to effective population size, and c refers to ploidy of the genome (*Li and Stephens, 2003*). These results do not indicate an obvious role for recombination in generating the derived haplotypes.

All haplotypes carrying putative resistance mutations in the *sdhB* gene arose in 2020–2021, along with all haplotypes of the *sdhD* gene apart from H6, H7, H16, and H23 (*Figure 4d*, *Supplementary file 5*). By 2024, the *sdhB* H6 haplotype became the most prevalent haplotype, while the non-resistant *sdhB* H3 haplotype was almost lost despite it being prevalent between 2013 and 2021 (*Figure 4d*, *Supplementary file 5*). Some individual populations had multiple haplotypes of the same putative resistance mutation (*Figure 4d*), and different haplotypes of the same mutation were often found in adjacent populations (*Figure 4a and d*, *Supplementary file 5*). In the ZJXS1 population used to produce the LabR and LabS strains, four haplotypes of the *sdhB* gene (H1, H3, H4, and H7) and six haplotypes of the *sdhD* gene (H1-H4, H11, and H24) were found, none of which had putative resistance mutations. After 66 generations of cyetpyrafen selection, two resistant haplotypes were fixed in the LabR population: H6 of *sdhB* and H10 of *sdhD*, which were likely derived from H1 of *sdhB* and H2 of *sdhD*, respectively (*Supplementary file 5*). These results illustrate the rapid and geographically widespread emergence of resistant alleles in these populations, linked to several mutations found in a range of different haplotypes.

## Single mutations have a large effect on resistance

We next tested resistance of homozygous individuals carrying mutations singly or in combination with other mutations. We first tested the population-level association between allele frequencies of the mutations and levels of resistance, using the five mutations present in at least ten populations (I260T, I260V, D116G, R119C, and R119L) and resistance indicated by survival at 1000 mg/L cyetpyrafen. When each mutation was analyzed separately, a weak association was found between survival and the frequency of the mutant allele (*Supplementary file 6*, linear regression $R^2$=0.091–0.303). The multivariate regression analysis revealed that frequency of three mutations, I260T (p=0.00128), I260V (p=0.00423), and D116G (p=0.00058), was significantly correlated with resistance of the field populations. When we considered the frequency of individuals having any of the five mutations, we found a strong positive association between survival and the frequency of the predominant allele ($R^2$=0.707), the frequency of individuals carrying at least one resistant allele (heterozygous and homozygous mutant genotypes; $R^2$=0.747), and the frequency of individuals with homozygous mutant genotypes ($R^2$=0.606; *Figure 5a–c*). These results showed that single mutations explained a low proportion of resistance variation across populations. But when considering all five mutations together, a substantial proportion of resistance variation was explained.

Homozygous lines at the mutation site(s) were created and tested for susceptibility to cyetpyrafen (*Figure 5d*). These lines carried homozygous I260V, D116G, R119E, R119L, and I260V+R119 C mutations. When treated with 1000 mg/L cyetpyrafen, mortality of the susceptible line was 100%, but was lower than 25.2% in all other lines, indicating that each mutation and a combination of two mutations conferred high resistance to cyetpyrafen. These results demonstrate that cyetpyrafen resistance is conferred by multiple large-effect mutations in the *sdhB* and *sdhD* genes, although further functional validations are needed.

## Discussion

Pesticide resistance can evolve rapidly, in contrast to traditional views of adaptation, which often emphasize a gradual process (*Fisher, 1930*). Due to the time taken for resistance to be recognized, evolutionary studies of pesticide resistance are often *post hoc*. Here, we conducted field monitoring of resistance to a novel acaricide, cyetpyrafen, following its 2017 release, with the opportunity to track resistance as it develops. We provided a longitudinal picture of resistance development in the field involving multiple and recurrently evolved large-effect mutations. Cross-resistance was found between cyetpyrafen and two other SDHi acaricides—cyflumetofen and cyenopyrafen—which were released in China before and after cyetpyrafen, respectively. Although we could not clearly identify the contributions of these acaricides to resistance evolution, they have only held a very small market share compared to cyetpyrafen, which is expected to be the main contributor to rapid resistance evolution.

Currently available pesticides mainly act by direct binding to receptors and enzymes, resulting in the inhibition of physiological functions associated with the nervous system, energy production system, growth and development, and midgut absorption (*Sparks et al., 2021*). Mutations that alter the binding kinetics of pesticides on these proteins and thus potentially confer pesticide resistance have been widely identified in agricultural pests and disease vectors (*De Rouck et al., 2023*; *Acford-Palmer et al., 2023*). Resistant mutations of some target genes are highly conserved both among populations and among species, perhaps due to the functional constraints of these targets (*ffrench-Constant et al., 1993b*; *Hawkins et al., 2019*). However, multiple mutations on single pesticide targets have also been identified in the same pest species and among different species. These cases often involve the voltage-gated sodium channel (VGSC) gene, where at least 5 and 10 mutations have been cumulatively reported across studies on TSSM and the ectoparasite *Dermanyssus gallinae*, respectively (*De Rouck et al., 2023*). Similarly, the evolution of resistance to acetolactate synthase (ALS)-inhibiting herbicides exhibits high repeatability across weed species, primarily driven by numerous SNPs within the ALS target gene (*Gaines et al., 2020*). To date, 29 distinct target-site mutations have been identified on eight amino acid residues across 67 weed species (*Tranel et al., 2025*). Notably, even within a single weed species, up to eight different mutations can occur on three amino acid residues (*Tranel et al., 2025*).

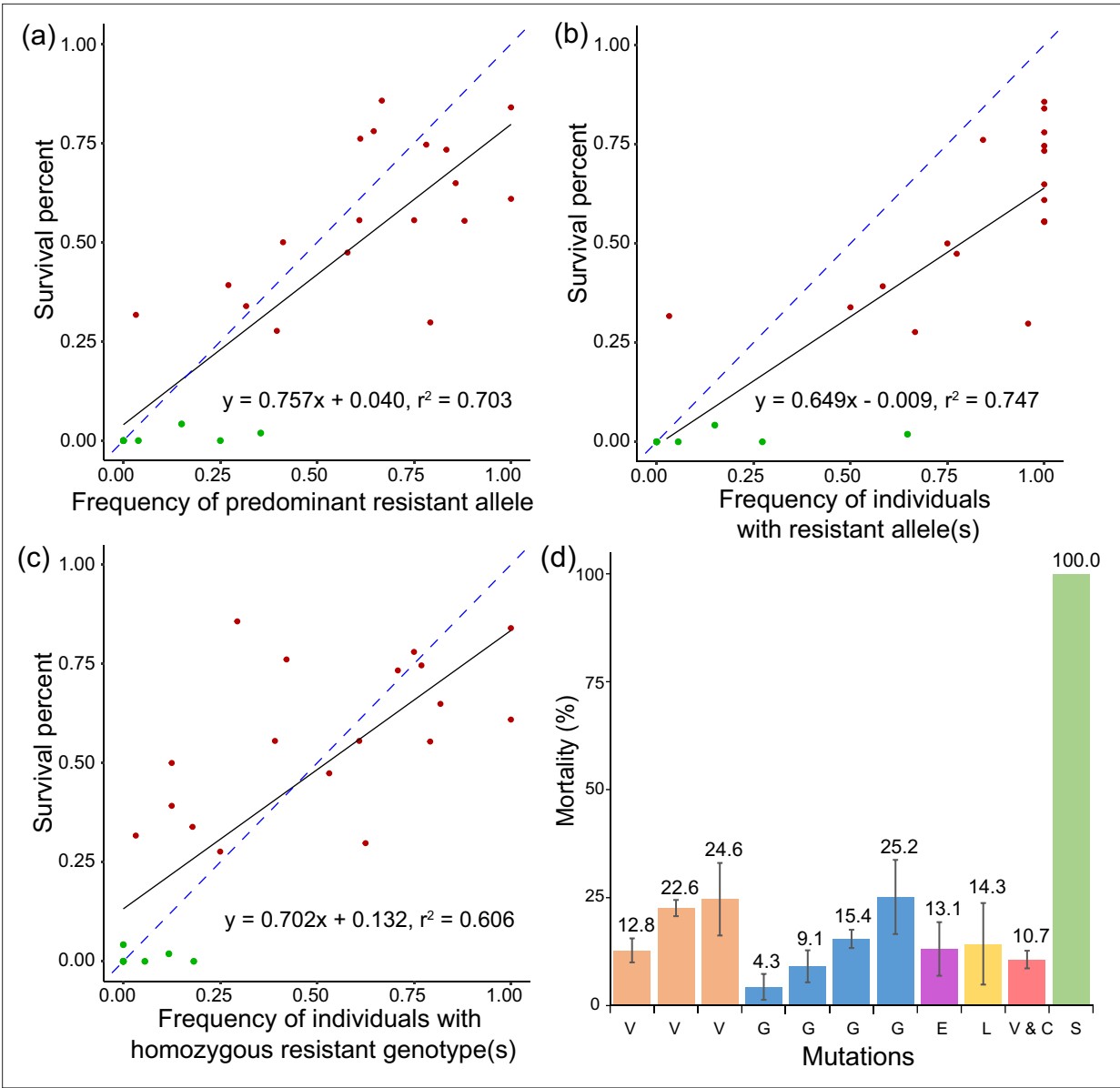

**Figure 5.** Function of target-site mutations in the *sdhB* and *sdhD* genes and survival of two-spotted spider mite *Tetranychus urticae* in cyetpyrafen resistance. (**a–c**) Association between frequency of target-site mutations in the *sdhB* and *sdhD* genes and survival of two-spotted spider mite *T. urticae* exposed to 1000 mg/L of cyetpyrafen. (**a**) Allele frequencies are measured as the predominant mutation in the population at either the *sdhB* or *sdhD* genes. (**b**) Allele frequency is the proportion of individuals carrying a resistant allele at either the *sdhB* or *sdhD* genes. (**c**) Allele frequency is the proportion of individuals carrying a homozygous genotype for either the *sdhB* or *sdhD* genes. Red and dark green points represent resistant and susceptible populations, respectively. Blue dashed lines indicate the expected association between two variables, i.e. *y=x*. (**d**) Mortality of mite lines carrying one of the homozygous mutations or a combination of two mutations exposed to 1000 mg/L of cyetpyrafen. V, I260V; G, D116G; E, R119E; L, R119L; V and C, I260V and R119C; S, susceptible populations LabS. Populations with the same homozygous mutations are shown in the same color. Error bars show standard error of the mean.

Here, we found 15 mutations on eight amino acid residues in TSSM that all arose within a few years following the introduction of cyetpyrafen. Notably, five of the substitutions involve R119 of the *sdhD* gene. As far as we know, this is the highest number of both mutations and altered residues ever found in a single pesticide target arthropod pests, all emerging in a short period following initial detection of resistance. These mutations were located at the Q-site of SDH targeted by acaricidal inhibitors of SDH. Among these mutations, I260T was previously found in a cyflumetofen-resistant population (*Sugimoto et al., 2020*), while I260V (with S56L in subunit C) was found in a pyflubumide and cyenopyrafen-resistant population (*Maeoka and Osakabe, 2021*). R119L has previously been

reported in cyflumetofen-resistant and -susceptible strains of TSSM but had not yet been validated as having a role in resistance (*Sugimoto et al., 2020*). These results provide information on the nature of novel mutations in a pesticide target and the high level of resistance to cyetpyrafen that they can confer. It would be interesting to further validate the impacts of the mutations across chemicals to establish patterns of cross-resistance given that the same mutation may confer a different level of resistance to different chemicals (*Maeoka and Osakabe, 2021*).

Our temporal collections of TSSM preceding cyetpyrafen release in the field indicated that the resistant mutations were not at an appreciable frequency prior to pesticide selection. Sequencing of 13 populations (involving a total of 2,317 TSSMs) from 2013 to 2018 did not recover any of the identified mutations. This points to a possible role of recurrent de novo mutation, although resistant alleles could have been present at very low frequencies before pesticide selection, which can be difficult to detect (*Hawkins et al., 2019*). In laboratory populations of TSSM, the H258Y mutation arose de novo and was subsequently favored by continuous selection, resulting in resistance to pyflubumide and cyenopyrafen (*Njiru et al., 2023*). Our selection experiments also led to new mutations (I260V, R119C) arising over 66 generations (2 years) of cyetpyrafen selection in the LabR population. Although it is hard to completely exclude the possibility of contamination during laboratory selection, because TSSM is tiny and can easily be dispersed passively, the haplotype analysis showed that the two mutations in the lab-selected population likely arose de novo. In the same way, multiple de novo mutations could have contributed to the rapid evolution of resistance in different populations and regions as documented in our field surveys.

Although few studies have traced resistance evolution in the field, they suggest that both de novo mutations and standing genetic variation can contribute to resistance, with their relative importance depending on the specific pesticides and pests involved (*Hawkins et al., 2019*). Resistant alleles can originate from a single event and spread to distant populations by migration (*Raymond et al., 1991*; *Daborn et al., 2002*), but resistant mutations can also have multiple origins (*Guan et al., 2021*; *Ilias et al., 2014*; *Troczka et al., 2012*; *Kreiner et al., 2022b*; *Schmidt et al., 2024*). Direct evidence for multiple origins comes from the presence of the same mutation on multiple haplotypes, assuming that recombination can be excluded as an explanation because insufficient time has elapsed after a resistant mutation is introduced into a new background (*Semenov et al., 2019*). In our study, the requisite recombinant haplotypes were rarely found in the same population, and we therefore suspect that recombination is unlikely to be the source of patterns of haplotype variation over short DNA sequences. We suspect that populations of mites like TSSM have a strong evolutionary potential to evolve in situ rather than relying on gene flow, and that there is a repeatability of evolutionary changes when an organism is exposed to the same selection pressure, a long-standing issue in evolutionary biology (*Konečná et al., 2021*). The recurrent appearance of the same mutation in different genetic backgrounds suggests that the likelihood of resistance evolution is high in TSSM and not dependent on gene flow, and the likelihood of this scenario is further increased by the fact that resistance can develop through several mutational options. We observed 2.7 resistant mutations per population in samples collected in 2024, seven years after the release of cyetpyrafen. The estimated mutation rate ($\Theta$) is 0.0193, assuming 20 generations per year for TSSM. An effective population size ($N_e$) of $2.29*10^6$ would be necessary to reach the number of de novo mutation observed in this study, given $\Theta = 3N_e\mu$ (haplodiploid sex determination of TSSM) and a mutation rate of $\mu = 2.8*10^{-9}$ per base pair per generation as estimated for *Drosophila melanogaster* (*Keightley et al., 2014*). The high reproductive capacity (>100 eggs per female) and short generation time of TSSM make it possible to reach such a large population size in the field.

Because field populations of TSSM can use multiple mutations to reach the same phenotypic endpoint (resistance), it is unlikely that all resistant mutations will be identified from laboratory selection studies, particularly if these involve a single laboratory population. Even in the case where multiple populations of TSSM have been selected for resistance (to cyenopyrafen and the related chemicals cyflumetofen and pyflubumide), QTL mapping still only identified a few of the relevant mutations providing resistance to each chemical (*Sugimoto et al., 2020*). Additionally, mutations identified in laboratory selection may not be involved in resistance in the field because of fitness costs. For instance, H258Y conferred resistance to cyenopyrafen and pyflubumide in laboratory-selected populations and is associated with a high fitness cost in TSSM (*Njiru et al., 2023*), and also likely reinforces cyflumetofen binding and toxicity (*Njiru et al., 2022*). Relying on a limited set of resistance

alleles identified in the laboratory for monitoring may bias inferences of resistance evolution in the field where many alleles can produce resistance. Genetic and evolutionary studies on field populations are then essential to develop a full picture of resistance evolution (*Konečná et al., 2021*; *Lucas et al., 2023*; *Schmidt et al., 2024*; *Kersten et al., 2023*; *Pezzini et al., 2024*).

Theoretical and empirical studies have shown that field-evolved resistance often involves large-effect alleles with a simple genetic basis, while laboratory selection may result in polygenic genetic architectures, due to differences in the occurrence of rare alleles (more likely in large field populations) and different selection pressures (more variable with intense periods in the field on occasions; *ffrench-Constant, 2013*; *Roush and McKenzie, 1987*; *Ffrench-Constant et al., 1990*; *McKenzie and Batterham, 1994*; *Crow, 1954*). This could potentially be mitigated by establishing laboratory selection lines using large starting populations from the field with pre-existing resistance alleles (*Gould et al., 1997*). Until recently, few studies have directly compared the genetic basis of resistance in laboratory populations with field-evolved resistance across a large geographical scale (*McKenzie et al., 1992*; *Ffrench-Constant et al., 1990*; *Crow, 1954*; *Legan et al., 2024*). Here, we identified large-effect alleles in both field- and laboratory-selected resistant populations. High-level resistance was generated after lab selection with cyetpyrafen for 33 of 65 generations, almost at the same rate as the field populations where high-level resistance was observed three years after the commercial release of cyetpyrafen. In field populations, it seems that multiple allelic options can be favored by selection due to parallel adaptation, and that the same allele may also end up residing in different haplotypes in different areas. Through migration, these alleles/backgrounds may eventually end up in the same population to produce a polyallelic basis of adaptation where particular alleles may then not reach high frequencies unless one haplotype is favored (*Ralph and Coop, 2010*).

Pesticide resistance provides many fascinating examples of rapid and recent evolution under high selection pressure, which can be leveraged to address fundamental issues about the evolutionary basis of adaptations to new environments (*Hawkins et al., 2019*), and also to examine the reliability of commonly used research methods. The development of high-throughput sequencing and population genomics methodologies provides an opportunity to detect selective signatures across the genome (*Van Etten et al., 2020*; *Cohen et al., 2022*; *Pélissié et al., 2022*). Although pesticide resistance is a rapid evolutionary process under strong selective pressures, identifying candidate genes directly from genomic signals remains difficult when adaptation occurs via soft sweeps (*Pennings and Hermisson, 2006*; *Schrider et al., 2015*). Soft sweeps may be common in adaptive evolution if the effective mutation rate ($4N_e\mu$)>>0.01, for example through large population sizes ($N_e$), and/or high allelic mutation rates (*Hermisson and Pennings, 2005*; *Pennings and Hermisson, 2006*). The multiple haplotypes and distinct evolutionary history of SDH genes among resistant field populations suggest a prevalence of soft selective sweeps through recurrent mutation as implicated in pesticide resistance in other pest systems, such as *Drosophila* (*Garud and Petrov, 2016*), and the oomycete *Plasmopara viticola* (*Delmas et al., 2017*). The haplotype diversity is likely to be underestimated, given that only two short fragments of *sdhB* and *sdhD* gene were used in this study. This is congruent with the 'messier' signals found in field populations relative to the artificially selected resistant laboratory population. Sweeps around the SDH genes are incomplete, or other contributing loci are undergoing selective sweeps, which may further reduce signal to detect adaptation (*Pennings and Hermisson, 2006*; *Schrider et al., 2015*). Fewer mutation types and haplotypes, along with higher mutation frequency, are more likely to be detected. Clearer signals of selective sweeps could emerge from haplotype-based linkage-disequilibrium methods using individual-level genomic data (*Hermisson and Pennings, 2017*).

We note a few caveats for our study. While we focused on target gene mutations, genes other than the sdh genes may play roles in the rapid resistance response. Environmental factors might influence population responses to cyetpyrafen and the structure of resistance alleles, but these were not examined. Incomplete dominance might also influence the expression of resistance, which could explain the decoupling of total sum frequency of resistance alleles from the percent survival (*Figure 2b* xii, *Figure 5b*). Future studies should incorporate functional genetic approaches to characterize allele dominance and environmental influence on resistance, using, for example, isogenic lines or CRISPR-Cas9 technology. However, the overall positive associations between resistance allele frequencies and survival post-cyetpyrafen exposure provide strong evidence that the sdh mutations observed in this study have a strong contributing role to resistance.

In conclusion, we have captured a detailed picture of contemporary evolutionary processes under pesticide selection in field populations of an invertebrate pest. We demonstrate that the rapid evolution of pesticide resistance is driven by an unprecedented number of population-specific mutations in target genes that arose recurrently. These findings reveal complex origins and diverse genetic bases of pesticide resistance in the field. These findings represent a challenge for SNP-specific molecular monitoring of resistance and for the implementation of effective resistance management strategies. They also highlight the value of amplicon and whole-genome sequencing to detect multiple target-site mutations. Finally, we call for future studies on field-evolved resistance to provide theoretical and practical insights into effective pest control, leveraging the genomics era.

# Materials and methods

## Samples

We used 72 field-collected and laboratory-selected populations of TSSM across China for bioassays, whole-genome resequencing, Sanger sequencing of target genes, and establishment of mutant strains (*Supplementary file 1*), including 15 populations collected before (2013–2017) and 55 populations collected after (2018–2024) the release of cyetpyrafen, as well as one laboratory-selected resistant population, and one laboratory-maintained susceptible population. More than 500 adults were collected from at least three points separated by 100 meters for each population. Samples were kept in 100% alcohol or transported live to the laboratory for chemical bioassays. Because TSSM has a haplodiploid sex-determination system, only female adults were randomly selected for bioassays and genotyping. An initial colony of about 4000 individuals was established from the field population ZJXS1 in 2018, which was subsequently divided into two parallel cohorts under controlled laboratory conditions. The selection group (LabR) was subjected to continuous selection pressure using cyetpyrafen, while the control group (LabS) was maintained under identical laboratory conditions without exposure to acaricidal agents. The TSSM strain was reared on French bean, *Pemphigus vulgaris* L., under 25 ± 1 °C, 60 ± 5% relative humidity and a photoperiod of L16h: D8h. We continuously selected this population in the laboratory by killing about 60–0% of randomly selected individuals every two generations, to obtain a laboratory-selected resistant strain. When the mortality was lower than 60%, another round of spraying was applied by increasing the dosage of cyetpyrafen. The $LC_{50}$ values were tested at $F_1$, $F_{32}$, $F_{54}$, $F_{60}$, $F_{62}$, and $F_{66}$ generations to quantify resistance development trajectories.

## Bioassay

Bioassays were conducted to test the susceptibility of TSSM to acaricides using a Potter Spray Tower (Burkard Scientific, London, UK). We used transparent plastic cups (6 cm in diameter and 4 cm in height) to keep the female adult mites. A layer of 0.2% agar was placed on the bottom of the cups to avoid leaves drying out. Fresh leaves of *P. vulgaris* were cut into a circle with a diameter of 6 cm to fit the cup and put onto the agar. The borders of the leaves were sealed using 0.2% agar. Thirty female adults were carefully moved into a plastic cup by suction trap. Two ml acaricide solution was applied through a Potter spray tower (Burkard Manufacturing Co. Ltd., UK) at 68.9 kPa and a sedimentation time of 30 s. Each population was exposed to between five and eight pre-estimated concentrations of acaricides, diluted with water containing 0.1% Triton X-100 (Beijing Solar BioScience and Technology Limited Company, China). The 0.1% Triton X-100 solution was used as a control treatment to correct the potential influence of adjuvants in the acaricides. The treated mites were kept under 25 °C, 60–70% relative humidity, and a photoperiod of 16 hr:8 hr (L:D). Mortality was assessed under a Stemi 305 stereomicroscope (Zeiss, Germany), 48 hr after spraying. The tested TSSMs were considered dead if no movement of appendages was observed when they were prodded with a fine brush at 24 hr and 48 hr after spraying. The 50% lethal concentration ($LC_{50}$) and its 95% confidence intervals were calculated using probit regressions implemented in the statistical software DPS v12.01 (*Tang and Zhang, 2013*).

## Pooled whole-genome resequencing and SNP calling

Between 50 and 400 individuals per population were used for pooled genome resequencing to obtain population genomic data for diversity estimates and demographic analyses. DNA libraries with insert sizes of 400 bp were constructed and sequenced on the Illumina NovaSeq 6000 platform to

obtain 150 bp paired-end reads. Raw reads were filtered using fastp v0.23.0 (*Chen et al., 2018*). The filtered reads were mapped to our recently assembled chromosome-level genome (*Cao et al., 2024*) using bwa-mem2 2.2.1 with default parameters (*Vasimuddin et al., 2019*). SAMtools v1.14 (*Li et al., 2009*) was used to sort the sequence data, removing duplicated reads, removing reads with mapping quality <20, and converting bam files to mpileup files. Indels in mpileup files were filtered out (with a 5 bp margin on each side) to avoid spurious SNP calling using scripts from PoPoolation (*Kofler et al., 2011a*). Then mpileup files were converted using scripts from PoPoolation2 (*Kofler et al., 2011b*).

## Population genetic diversity and demography analyses

Genetic diversity in each population was quantified by estimating nucleotide diversity ($\pi$) and Tajima's $D$ across 5 kb non-overlapping windows, using PoPoolation (*Kofler et al., 2011a*). To reduce the impact of read depth on estimates of nucleotide diversity, each alignment was converted into a population-specific pileup file using SAMtools, and reads were subsampled without replacement up to a depth of 50 (*Hoban et al., 2016*). The mean $F_{ST}$ across all windows for each pairwise population comparison was calculated using PoPoolation2 (*Kofler et al., 2011a*) across 5 kb non-overlapping windows. To infer population relationships and historical migration, a rooted maximum likelihood tree was constructed with a strict drift model (no migration) using Treemix v1.13 (*Pickrell and Pritchard, 2012*). The SCCD population was set as root and we tested migration events from 0 to 10 with at least 100 runs for each case. The best number of migration events was determined by the second-order rate of change in likelihood ($\Delta m$) across incremental migration events ($m$) using the Evanno method (*Evanno et al., 2005*) embedded in R package 'optM' (*Fisher, 2021*). For the best m, we performed 1000 runs to generate 1000 maximum likelihood trees and obtained a consensus tree using the program SumTrees version 4.10 (*Sukumaran and Holder, 2010*).

## Genome scan of outlier SNPs

We investigated the genomic signals of selection around the four SDH genes (±1 Mb) and across the whole genome. The main statistic used to identify outlier regions was the population branch excess (PBE) statistic (*Yassin et al., 2016*). The PBE statistic is based on comparisons of $F_{ST}$ values among three populations, comprising a focal population and two nonfocal populations (*Pool et al., 2017*). In the analysis, each resistant population was set as the focal population, and two susceptible populations (LabS and NMHH2) were set as the nonfocal populations. PBE values were calculated for 5 kb non-overlapping sliding windows. Windows with top 5% PBE value were considered outliers. Patterns of nucleotide diversity ($\pi$) estimates and Tajima's $D$ values around the SDH genes were also examined. We expected resistant populations to have lower nucleotide diversity and Tajima's $D$ values near the SDH genes due to directional selection favoring resistant haplotypes. The coding effects of SNPs within the SDH genes were obtained using SnpEff v4.3 (*Cingolani et al., 2012*).

## SDH homology modeling

The three-dimensional (3D) structures of the wild and mutational SDH from TSSM were modeled with SWISS-MODEL (*Waterhouse et al., 2018*) using 8GS8.1 (*Du et al., 2023*) as a template. To examine the conservation of *SdhB* and *SdhD* genes, we collected the protein sequences of two genes of different animals in the protein database of NCBI (https://www.ncbi.nlm.nih.gov/protein). Then, MUSCLE in MEGA v11 (*Tamura et al., 2021*) was used to perform sequence alignment and trim protein sequences around mutation points. After removing identical sequences of the same species, the multiple sequence alignment sequences were used to determine conservation and visualized with WebLogo 3 (*Crooks et al., 2004*).

## PCR amplification and sequencing of target genes

The target gene sequences of 16–48 individuals from each population were used to characterize the spatiotemporal variation of these target-site mutations in these populations before and after the release of cyetpyrafen (*Supplementary file 1*). PCR primers were designed for sequences of the *sdhB* gene (sdhb_F: GAGTCTAGCCGCAGTCTGAT, sdhb_R: ACTTGGGACCTGCTGTACTC) and the *sdhD* gene (sdhd_F: GCGATGGATAAGGCATAGACG, sdhd_R: TCTGGTCAGTGTGGTTCCTC) to capture the candidate mutation sites. Polymerase chain reaction (PCR) was conducted in the Mastercycler pro system (Eppendorf, Germany) under an annealing temperature of 54 °C for 1 min using Promega

Go Taq Master Mixes (Madison, Wisconsin, USA). Amplified products were purified and sequenced directly from one strand using an ABI 3730xl DNA Analyzer by Sangon Biotech Co. Ltd. (Shanghai, China).

## Haplotype analysis

Haplotypes of each gene were identified from Sanger sequencing. We constructed haplotype networks for *sdhB* and *sdhD* from individuals that contained homozygous resistant genotypes. Haplotype networks were constructed using the R package *pegas* (*Paradis, 2010*) and were visualized with the R package *ggplot2*.

## Association analysis between mutation frequency and resistance level

Regressions were run to assess the strength of association between target-site mutation frequencies and mean population-level resistance phenotypes. Since $LC_{50}$ of some populations could not be calculated or were inaccurate due to high resistance levels, the survival rate after spraying with 1000 mg/L cyetpyrafen, which is about 6.7–10-fold of field-recommended dose, was used to quantify the resistance level of TSSM to cyetpyrafen. Because multiple target-site mutations existed, we evaluated the associations using different metrics of target-site mutation frequencies: (i) the frequency of the predominant mutation in each population, (ii) the frequency of individuals carrying any mutation in each population, and (iii) the frequency of individuals with homozygous mutations in each population. Regressions were fitted as linear models using the *lm* function from the R package *stats* and visualized with *ggplot2*. We did not adjust associations for relatedness among populations because the evolutionary events leading to resistance appear to be largely independent. A logistic regression model was used to track trajectories of allele frequencies. The selection coefficient of each allele (s) and their joint effect ($\bar{s}$) were estimated (*Kreiner et al., 2022a*). We calculated $F_{ST}$ using the frequency of resistance alleles following *Hartl and Clark, 1997* and explored isolation by distance (IBD) with Mantel tests in the R package *vegan*.

## Establishment of homozygous lines

Single immature TSSM females were picked from resistant populations and reared individually in a petri dish. When unmated, females will only produce haploid male eggs. To obtain the genotype of males for the *sdh* genes, the DNA of each female was extracted and genotyped as mentioned above after it had laid more than 10 eggs, while the eggs (>10) were allowed to develop into male adults ($F_1$ male adults). To obtain the genotype of virgin females, the first six eggs were genotyped while the females were kept live ($F_0$ females). Then, each pair of $F_0$ female and $F_1$ male adults with the same homozygous genotypes for the six mutation sites was placed into the same petri dish and left to mate and produce male and female progeny, with females from the line generated defined as a line homozygous for the target resistance mutation(s). The mated male and female pairs were from the same population to reduce differences in their genetic backgrounds. The homozygous lines at the mutation sites were tested for susceptibility to cyetpyrafen with the bioassay method described above.

## Acknowledgements

We thank Ming-Liang Li, Hong-Ping Tian, Liang-Bin Zeng, for their assistance with sample collections, Prof. Yi-Dong Wu (Nanjing Agricultural University) and Prof. Thomas Van Leeuwen (Ghent University) for their comments towards improving our manuscript. This study is supported by the National Key R&D Program of China (2023YFD1401200), Program of Beijing Academy of Agriculture and Forestry Sciences (QNJJ202006, JKZX202208), and the Grains Research and Development Corporation and Horticulture Innovation (Australia).

## Additional information

### Funding

| Funder | Grant reference number | Author |
| --- | --- | --- |
| National Key Research and Development Program of China | 2023YFD1401200 | Shu-Jun Wei |
| Program of Beijing Academy of Agriculture and Forestry Sciences | QNJJ202006 | Jin-Cui Chen |
| Program of Beijing Academy of Agriculture and Forestry Sciences | JKZX202208 | Shu-Jun Wei |
| Grains Research and Development Corporation | | Joshua A Thia |
| Horticulture Innovation Australia Limited | | Joshua A Thia |

The funders had no role in study design, data collection and interpretation, or the decision to submit the work for publication.

### Author contributions

Li-Jun Cao, Data curation, Software, Formal analysis, Investigation, Visualization, Methodology, Writing – original draft; Jin-Cui Chen, Resources, Data curation; Joshua A Thia, Writing – original draft; Thomas L Schmidt, Richard Ffrench-Constant, Qiong Yang, Ya-Jun Gong, Hu Li, Xuexin Chen, Writing – review and editing; Lin-Xi Zhang, Yu Yang, Meng-Chu Yuan, Jia-Yue Zhang, Xiao-Yang Zhang, Investigation; Ary A Hoffmann, Supervision, Funding acquisition, Writing – review and editing; Shu-Jun Wei, Conceptualization, Resources, Supervision, Funding acquisition, Validation, Methodology, Writing – original draft, Writing – review and editing

### Author ORCIDs

Li-Jun Cao https://orcid.org/0000-0002-4595-0136
Joshua A Thia https://orcid.org/0000-0001-9084-0959
Qiong Yang https://orcid.org/0000-0003-3336-5703
Ary A Hoffmann https://orcid.org/0000-0001-9497-7645
Shu-Jun Wei https://orcid.org/0000-0001-7398-0968

Reviewer #1 (Public review): https://doi.org/10.7554/eLife.106288.3.sa1
Reviewer #2 (Public review): https://doi.org/10.7554/eLife.106288.3.sa2
Author response https://doi.org/10.7554/eLife.106288.3.sa3

## Additional files

### Supplementary files
MDAR checklist

Supplementary file 1. Population information of the two-spotted spider mite *Tetranychus urticae* used in this study.

Supplementary file 2. Mutant allele frequency (Sanger / pooled sequencing) and susceptibility of the two-spotted spider mites *Tetranychus urticae* to cyetpyrafen.

Supplementary file 3. Laboratory selection and cross-resistance of *Tetranychus urticae*.

Supplementary file 4. Sequencing information and genetic diversity of 22 populations of the two-spotted spider mite *Tetranychus urticae*.

Supplementary file 5. Distribution of haplotypes carrying amino acid mutations in populations.

Supplementary file 6. Correlation between eight mutations on SDH genes and level of resistance (survival percentages under 1000 mg/L).

## Data availability

Raw reads of whole genome resequencing data were submitted to NCBI Sequence Read Archive, with accession numbers SRR30666541 - SRR30666561 and SRR28000465. All scripts and data used in the study were deposited to GitHub: https://github.com/caolijun/TU_pool-seq (copy archived at *Cao, 2025*) and Figshare: https://doi.org/10.6084/m9.figshare.27231003.v3.

The following datasets were generated:

| Author(s) | Year | Dataset title | Dataset URL | Database and Identifier |
| --- | --- | --- | --- | --- |
| Wei SJ, Cao LJ | 2024 | Recurrent Mutations Drive the Rapid Evolution of Pesticide Resistance in the Two-spotted Spider Mite Tetranychus urticae | https://doi.org/10.6084/m9.figshare.27231003.v3 | figshare, 10.6084/m9.figshare.27231003.v3 |
| Cao LJ | 2024 | WGS of Tetranychus urticae: NMHH2 | https://www.ncbi.nlm.nih.gov/sra/?term=SRR30666541 | NCBI Sequence Read Archive, SRR30666541 |
| Cao LJ | 2024 | WGS of Tetranychus urticae: SDSG3 | https://www.ncbi.nlm.nih.gov/sra/?term=SRR30666542 | NCBI Sequence Read Archive, SRR30666542 |
| Cao LJ | 2024 | WGS of Tetranychus urticae: JXNC | https://www.ncbi.nlm.nih.gov/sra/?term=SRR30666543 | NCBI Sequence Read Archive, SRR30666543 |
| Cao LJ | 2024 | WGS of Tetranychus urticae: HNCS1 | https://www.ncbi.nlm.nih.gov/sra/?term=SRR30666544 | NCBI Sequence Read Archive, SRR30666544 |
| Cao LJ | 2024 | WGS of Tetranychus urticae: SHPD | https://www.ncbi.nlm.nih.gov/sra/?term=SRR30666545 | NCBI Sequence Read Archive, SRR30666545 |
| Cao LJ | 2024 | WGS of Tetranychus urticae: HNHK | https://www.ncbi.nlm.nih.gov/sra/?term=SRR30666546 | NCBI Sequence Read Archive, SRR30666546 |
| Cao LJ | 2024 | WGS of Tetranychus urticae: SCCD1 | https://www.ncbi.nlm.nih.gov/sra/?term=SRR30666547 | NCBI Sequence Read Archive, SRR30666547 |
| Cao LJ | 2024 | WGS of Tetranychus urticae: LabS | https://www.ncbi.nlm.nih.gov/sra/?term=SRR30666548 | NCBI Sequence Read Archive, SRR30666548 |
| Cao LJ | 2024 | WGS of Tetranychus urticae: ZJHZ2 | https://www.ncbi.nlm.nih.gov/sra/?term=SRR30666550 | NCBI Sequence Read Archive, SRR30666550 |
| Cao LJ | 2024 | WGS of Tetranychus urticae: ZJHZ1 | https://www.ncbi.nlm.nih.gov/sra/?term=SRR30666551 | NCBI Sequence Read Archive, SRR30666551 |
| Cao LJ | 2024 | WGS of Tetranychus urticae: YNYX | https://www.ncbi.nlm.nih.gov/sra/?term=SRR30666552 | NCBI Sequence Read Archive, SRR30666552 |
| Cao LJ | 2024 | WGS of Tetranychus urticae: QHHD | https://www.ncbi.nlm.nih.gov/sra/?term=SRR30666553 | NCBI Sequence Read Archive, SRR30666553 |
| Cao LJ | 2024 | WGS of Tetranychus urticae: LNDD | https://www.ncbi.nlm.nih.gov/sra/?term=SRR30666554 | NCBI Sequence Read Archive, SRR30666554 |
| Cao LJ | 2024 | WGS of Tetranychus urticae: HNCS2 | https://www.ncbi.nlm.nih.gov/sra/?term=SRR30666555 | NCBI Sequence Read Archive, SRR30666555 |

*Continued on next page*

*Continued*

| Author(s) | Year | Dataset title | Dataset URL | Database and Identifier |
|---|---|---|---|---|
| Cao LJ | 2024 | WGS of Tetranychus urticae: GXNN | https://www.ncbi.nlm.nih.gov/sra/?term=SRR30666556 | NCBI Sequence Read Archive, SRR30666556 |
| Cao LJ | 2024 | WGS of Tetranychus urticae: BJDX6 | https://www.ncbi.nlm.nih.gov/sra/?term=SRR30666557 | NCBI Sequence Read Archive, SRR30666557 |
| Cao LJ | 2024 | WGS of Tetranychus urticae: SDQZ | https://www.ncbi.nlm.nih.gov/sra/?term=SRR30666558 | NCBI Sequence Read Archive, SRR30666558 |
| Cao LJ | 2024 | WGS of Tetranychus urticae: BJCP4 | https://www.ncbi.nlm.nih.gov/sra/?term=SRR30666559 | NCBI Sequence Read Archive, SRR30666559 |
| Cao LJ | 2024 | WGS of Tetranychus urticae: AHHN | https://www.ncbi.nlm.nih.gov/sra/?term=SRR30666560 | NCBI Sequence Read Archive, SRR30666560 |
| Cao LJ | 2024 | WGS of Tetranychus urticae: SXYQ | https://www.ncbi.nlm.nih.gov/sra/?term=SRR30666561 | NCBI Sequence Read Archive, SRR30666561 |
| Cao LJ | 2024 | Tetranychus urticae isolate:ZJXS2018_labR Genome sequencing | https://www.ncbi.nlm.nih.gov/sra/?term=SRR28000465 | NCBI Sequence Read Archive, SRR28000465 |
| Cao LJ | 2024 | WGS of Tetranychus urticae: SDRZNCBI Sequence Read Archive | https://www.ncbi.nlm.nih.gov/sra/?term=SRR30666549 | NCBI Sequence Read Archive, SRR30666549 |

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
