## [Editor Report · eLife Assessment]

This study provides **important** insights into the evolution of pesticide resistance, demonstrating that resistance can arise rapidly and repeatedly, which complements prior work on parallel evolution across species. The combination of extensive temporal sampling in the field, experimental evolution, and genomics makes for **compelling** findings. The authors are to be commended for acknowledging the main limitations of their study in the Discussion. Framing the work in a broader context of resistance beyond arthropod pests would further increase the appeal of the study, which is of relevance for both agronomic practitioners and evolutionary biologists.

---

## [Referee Report · Reviewer #1 (Public review)]

Summary:

The study by Cao et al. provides a compelling investigation into the role of mutational input in the rapid evolution of pesticide resistance, focusing on the two-spotted spider mite's response to the recent introduction of the acaricide cyetpyrafen. This well-documented introduction of the pesticide-and thus a clearly defined history of selection-offers a powerful framework for studying the temporal dynamics of rapid adaptation. The authors combine resistance phenotyping across multiple populations, extensive resequencing to track the frequency of resistance alleles, and genomic analyses of selection in both contemporary and historical samples. These approaches are further complemented by laboratory-based experimental evolution, which serves as a baseline for understanding the genetic architecture of resistance across mite populations in China. Their analyses identify two key resistance-associated genes, sdhB and sdhD, within which they detect 15 mutations in wild-collected samples. Protein modeling reveals that these mutations cluster around the pesticide's binding site, suggesting a direct functional role in resistance. The authors further examine signatures of selective sweeps and their distribution across populations to infer the mechanisms-such as de novo mutation or gene flow-driving the spread of resistance, a crucial consideration for predicting evolutionary responses to extreme selection pressure. Overall, this is a well-rounded, thoughtfully designed and well-written manuscript. It shows significant novelty, as it is relatively rare to integrate broad-scale evolutionary inference from natural populations with experimentally informed bioassays, however, follow up work will be needed to fully resolve haplotype structure and the functional effects of resistance mutations in the system.

Strengths:

One of the most compelling aspects of this study is its integration of genomic time-series data in natural populations with controlled experimental evolution. By coupling genome sequencing of resistant field populations with laboratory selection experiments, the authors tease apart the individual effects of resistance alleles along with regions of the genome where selection is expected to occur, and compare that to the observed frequency in the wild populations over space and time. Their temporal data clearly demonstrates the pace at which evolution can occur in response to extreme selection. This type of approach is a powerful roadmap for the rest of the field of rapid adaptation.

The study effectively links specific genetic changes to resistance phenotypes. The identification of sdhB and sdhD mutations as major drivers of cyetpyrafen resistance is well supported by allele frequency shifts in both field and experimental populations. The scope of their sampling clearly facilitated the remarkable number of observed mutations within these target genes, and the authors provide a careful discussion of the likelihood of these mutations from de novo or standing variation. Furthermore, the discovered cross-resistance that these mutations confer to other mitochondrial complex II inhibitors highlights the potential for broader resistance management and evolution.

Weaknesses:

(1) Pleiotropy without pesticide modes of action (cyflumetofen and cyetpyrafen) may also play a role in the rapid response to the focal pesticide in this study

(2) Other aspects of the environment that might influence selection were not considered in the structure of resistance alleles (i.e. climate, elevation)

(3) Very little data were used for haplotype reconstruction, only 8 SNPs, and this excluded all heterozygous alleles, which could dramatically influence the complexity of these inferred haplotype networks.

(4) Single Mutations and Their Effects:

- Allelic effects were not estimated in isogenic lines, so the effects presented also include heterogeneity from allelic interactions with the genomic background

- The authors see populations that segregate for resistance mutations but that have no survival to pesticides. This suggests either not all of the resistance mutations studied here actually have functional effects or that dominance is playing a role in masking their effects in the heterozygous state.

---

## [Referee Report · Reviewer #2 (Public review)]

Summary:

This paper investigates the evolution of pesticide resistance in the two-spotted spider mite following the introduction of an SDHI acaricide, cyatpyrafen, in China. The authors make use of cyatpyrafen-naive populations collected before that pesticide was first used, as well as more recent populations (both sensitive and resistant) to conduct comparative population genomics. They report 15 different mutations in the insecticide target site from resistant populations, many reported here for the first time, and look at the mutation and selection processes underlying the evolution of resistance, through GWAS, haplotype mapping, and testing for loss of diversity indicating selective sweeps. None of the target site mutations found in resistant populations was found in pre-exposure populations, suggesting that the mutations may have arisen de novo rather than being present as standing variation, unless initially present at very low frequencies; a de novo origin is also supported by evidence of selective sweeps in some resistant populations. Furthermore, there is no significant evidence of migration of resistant genotypes between the sampled field populations indicating multiple origins of common mutations. Overall, this indicates a very high mutation rate and a wide range of mutational pathways to resistance for this target site in this pest species. The series of population genomic analyses carried out here, in addition to the evolutionary processes that appear to underly resistance development in this case, could have implications for the study of resistance evolution more widely.

Strengths:

This paper combines phenotypic characterisation with extensive comparative population genomics, made possible by the availability of multiple population samples (each with hundreds of individuals) collected before as well as after then introduction of the pesticide cyatpyrafen, as well as lab-evolved lines. This resuts in findings of mutation and selection processes that can be related back to the pesticide resistance trait of concern. Large numbers of mites were tested phenotypically to show the levels of resistance present, and the authors also made near-isogenic lines to confirm the phenotypic effects of key mutations. The population genomic analyses consider a range of alternative hypotheses, including mutations arising by de novo mutation or selection from standing genetic variation; and mutations in different populations arising independently or arriving by migration. The claim that mutations most likley arose by multiple repeated de novo mutations is therefore supported by multiple lines of evidence: the direct evidence of none of the mutations being found in over 2000 individuals from naive populations, and the indirect evidence from population genomics showing evidence of selective sweeps but not of significant migration between the sampled populations.

Weaknesses:

As acknowledged within the discussion, whilst evidence supports a de novo origin of the resistance associated mutations, this cannot be proven definitively as mutations may have been present at a very low frequency and therefore not found within the tested pesticide-naive population samples.

Near-isofemale lines were made to confirm the resistance levels associated with five of the 15 mutations, but otherwise the genotype-phenotype associations are correlative as confirmation by functional genetics was beyond the scope of this study.

Comments on revisions:

My recommendations have all been addressed in the revised version.

---

## [Author Response]

The following is the authors’ response to the original reviews

**Public Reviews:**

**Reviewer #1 (Public review):**
Summary:The study by Cao et al. provides a compelling investigation into the role of mutational input in the rapid evolution of pesticide resistance, focusing on the two-spotted spider mite's response to the recent introduction of the acaricide cyetpyrafen. This well-documented introduction of the pesticide - and thus a clearly defined history of selection - offers a powerful framework for studying the temporal dynamics of rapid adaptation. The authors combine resistance phenotyping across multiple populations, extensive resequencing to track the frequency of resistance alleles, and genomic analyses of selection in both contemporary and historical samples. These approaches are further complemented by laboratory-based experimental evolution, which serves as a baseline for understanding the genetic architecture of resistance across mite populations in China. Their analyses identify two key resistance-associated genes, sdhB and sdhD, within which they detect 15 mutations in wild-collected samples. Protein modeling reveals that these mutations cluster around the pesticide's binding site, suggesting a direct functional role in resistance. The authors further examine signatures of selective sweeps and their distribution across populations to infer the mechanisms - such as de novo mutation or gene flow-driving the spread of resistance, a crucial consideration for predicting evolutionary responses to extreme selection pressure. Overall, this is a well-rounded, thoughtfully designed, and well-written manuscript. It shows significant novelty, as it is relatively rare to integrate broad-scale evolutionary inference from natural populations with experimentally informed bioassays, however, some aspects of the methods and discussion have an opportunity to be clarified and strengthened.Strengths:One of the most compelling aspects of this study is its integration of genomic time-series data in natural populations with controlled experimental evolution. By coupling genome sequencing of resistant field populations with laboratory selection experiments, the authors tease apart the individual effects of resistance alleles along with regions of the genome where selection is expected to occur, and compare that to the observed frequency in the wild populations over space and time. Their temporal data clearly demonstrates the pace at which evolution can occur in response to extreme selection. This type of approach is a powerful roadmap for the rest of the field of rapid adaptation.The study effectively links specific genetic changes to resistance phenotypes. The identification of sdhB and sdhD mutations as major drivers of cyetpyrafen resistance is well-supported by allele frequency shifts in both field and experimental populations. The scope of their sampling clearly facilitated the remarkable number of observed mutations within these target genes, and the authors provide a careful discussion of the likelihood of these mutations from de novo or standing variation. Furthermore, the discovered cross-resistance that these mutations confer to other mitochondrial complex II inhibitors highlights the potential for broader resistance management and evolution.Weaknesses:(1) Experimental Evolution:- Additional information about the lab experimental evolution would be useful in the main text. Specifically, the dose of cyetpyrafen used should be clarified, especially with respect to the LD50 values. How does it compare to recommended field doses? This is expected to influence the architecture of resistance evolution. What was the sample size? This will help readers contextualize how the experimental design could influence the role of standing variation.

The experimental design involved sampling approximately 6,000 individuals from the wild population ZJSX1, which were subsequently divided into two parallel cohorts under controlled laboratory conditions. The selection group (LabR) was subjected to continuous selection pressure using cyetpyrafen, while the control group (LabS) was maintained under identical laboratory conditions without exposure to acyetpyrafen. A dynamic selection regime was implemented wherein the acaricide dosage was systematically adjusted every two generations to maintain a consistent selection intensity, achieving a mortality rate of 60% ± 10% in the LabR population. This adaptive dosage strategy ensured sustained evolutionary pressure while preventing population collapse. The LC_50_ values were tested at F1, F32, F54, F60, F62, and F66 generations using standardized bioassay protocols to quantify resistance development trajectories and optimize dosage for subsequent selection cycles. We provided the additional information in subsection 4.1 of the materials and methods section.

- The finding that lab-evolved strains show cross-resistance is interesting, but potentially complicates the story. It would help to know more about the other mitochondrial complex II inhibitors used across China and their impact on adaptive dynamics at these loci, particularly regarding pre-existing resistance alleles. For example, a comparison of usage data from 2013, 2017, and 2019 could help explain whether cyetpyrafen was the main driver of resistance or if previous pesticides played a role. What happened in 2020 that caused such rapid evolution 3 years after launch?

Although the introduction of the other two SDHI acaricides complicates the story, we would like to provide a complete background on the usage of acaricides with this mode of action in China. Although cyflumetofen was released in 2013 before cyetpyrafen, and cyenopyrafen was released in 2019 after cyetpyrafen, their market share is minor (about 3.2%) compared to cyetpyrafen (about 96.8%, personal communication). Since cross-resistance is reported among SDHIs, we could not exclude the contribution of cyflumetofen to the initial accumulation of resistance alleles, but the effect should be minor, both because of their minimal market share and because of the independent evolution of resistance in the field as found in our study. Although the contribution of cyflumetofen and cyenopyrafen cannot be entirely excluded, the rapid evolution of resistance seems likely to be mainly explained by the intensive application of cyetpyrafen. To clarify this issue, we added relevant information in the first paragraph of the discussion section.

(2) Evolutionary history of resistance alleles:- It would be beneficial to examine the population structure of the sampled populations, especially regarding the role of migration. Though resistance evolution appears to have had minimal impact on genome-wide diversity (as shown in Supplementary Figure 2), could admixture be influencing the results? An explicit multivariate regression framework could help to understand factors influencing diversity across populations, as right now much is left to the readers' visual acuity.

The genetic structure of the populations was examined by Treemix analysis. We detected only one migration event from JXNC to SHPD (no resistance data available for these two populations), suggesting a limited role for migration to resistance evolution. The multiple regression analysis revealed that overall genetic diversity and Tajima’s *D* across the genome were not significantly associated with resistance levels, genetic structure or geographic coordinates (*P* > 0.05), which all support a limited role of migration in resistance development.

- It is unclear why lab populations were included in the migration/treemix analysis. We might suggest redoing the analysis without including the laboratory populations to reveal biologically plausible patterns of resistance evolution.

Thank you for the constructive suggestion. The Treemix analysis was redone by removing laboratory populations and is now reported.

- Can the authors explore isolation by distance (IBD) in the frequency of resistance alleles?

Thank you for the constructive suggestion. No significant isolation-by-distance pattern was detected for resistance allele frequencies across all surveyed years (2020: P=0.73; 2021: P=0.52; 2023: P=0.16; Mantel test). We added these results to the text.

- Given the claim regarding the novelty of the number of pesticide resistance mutations, it is important to acknowledge the evolution of resistance to all pesticides (antibiotics, herbicides, etc.). ALS-inhibiting herbicides have driven remarkable repeatability across species based on numerous SNPs within the target gene.

We appreciate this comment, which highlights the need to place our findings within the broader evolutionary context of pesticide resistance. We have investigated references relevant to the evolution of resistance to diverse pesticides. As far as we can tell, the 15 target mutations in eight amino acid residues are among the highest number of pesticide resistance mutations detected, especially within the context of animal studies. We have added relevant text to the second paragraph of the discussion.

- Figure 5 A-B. Why not run a multivariate regression with status at each resistance mutation encoded as a separate predictor? It is interesting that focusing on the predominant mutation gives the strongest r2, but it is somewhat unintuitive and masks some interesting variation among populations.

We conducted a multiple regression analysis to explore the influence of multiple mutations on resistance levels of field populations. However of 15 putative resistant mutations, only five were detected in more than three populations where bioassay data are available, i.e. I260T, I260V, D116G, R119C, R119L. The frequency of three of these mutations, I260T (P = 0.00128), I260V (P = 0.00423) and D116G (P = 0.00058), are significantly correlated with the resistance level of field populations. This has been added.

(3) Haplotype Reconstruction (Line 271-):- We are a bit sceptical of the methods taken to reconstruct these haplotypes. It seems as though the authors did so with Sanger sequencing (this should be mentioned in the text), focusing only on homozygous SNPs. How many such SNPs were used to reconstruct haplotypes, along what length of sequence? For how many individuals were haplotypes reconstructed? Nonetheless, I appreciated that the authors looked into the extent to which the reconstructed haplotypes could be driven by recombination. Can the authors elaborate on the calculations in line 296? Is that the census population size estimate or effective?

Because haplotypes could not be determined when more than two loci were heterozygous, we detected haplotypes from sequencing data with at most one heterozygous locus. In total 844 individuals and 696 individuals were used to detect haplotypes of sdhB and sdhD. We detected 11 haplotypes (with 8 SNPs) and 24 haplotypes (with 11 SNPs) along 216 bp of the sdhB and 155 bp of the sdhD genes, respectively. Please see the fifth paragraph of subsection 2.4. We used ρ = 4 × Ne × d (genetic distance) (Li and Stephens, 2003) to calculate the number of effective individuals for one recombination event.

(4) Single Mutations and Their Effect (line 312-):- It's not entirely clear how the breeding scheme resulted in near-isogenic lines. Could the authors provide a clearer explanation of the process and its biological implications?

To investigate the effect of single mutations or their combination on resistance levels, we isolated the females and males with the same homozygous/ hemizygous genotypes for creating homozygous lines. Females from these lines were not near-isogenic, but homozygous for the critical mutations. We revised the description in the methods section to clearly define these lines.

- If they are indeed isogenic, it's interesting that individual resistance mutations have effects on resistance that vary considerably among lines. Could the authors run a multivariate analysis including all potential resistance SNPs to account for interactions between them? Given the variable effects of the D116G substitution (ranging from 4-25%), could polygenic or epistatic factors be influencing the evolution of resistance?

We couldn’t conduct multivariate analysis because most lines have only one resistant SNP. The four lines homozygous for 116G were from the same population. The variable mortality may reflect other unknown mechanisms but these are beyond the scope of this study.

- Why are there some populations that segregate for resistance mutations but have no survival to pesticides (i.e., the green points in Figure 5)? Some discussion of this heterogeneity seems required in the absence of validation of the effects of these particular mutations. Could it be dominance playing a role, or do the authors have some other explanation?

We didn’t investigate the degree of dominance of each mutation. The mutation I260V shows incompletely dominant inheritance (Sun, et al. 2022). To investigate survival rate of different populations, the two-spotted spider mite *T. urticae* was exposed to 1000 mg/L of cyetpyrafen, higher than the recommended field dose of 100 mg/L. Such a high concentration may lead to death of an individual heterozygous for certain mutations, such as I260V.

- The authors mention that all resistance mutations co-localized to the Q-site. Is this where the pesticide binds? This seems like an important point to follow their argument for these being resistance-related.

Yes. We revised Fig. 3c to show the Q-site.

(5) Statistical Considerations for Allele Frequency Changes (Figure 3):- It might be helpful to use a logistic regression model to assess the rate of allele frequency changes and determine the strength of selection acting on these alleles (e.g., Kreiner et al. 2022; Patel et al. 2024). This approach could refine the interpretation of selection dynamics over time.

Thank you for this suggestion. A logistic regression model was used to track allele frequencies trajectories. The selection coefficient of each allele and their joint effects were estimated.

**Reviewer #2 (Public review):**
Summary:This paper investigates the evolution of pesticide resistance in the two-spotted spider mite following the introduction of an SDHI acaricide, cyatpyrafen, in China. The authors make use of cyatpyrafen-naive populations collected before that pesticide was first used, as well as more recent populations (both sensitive and resistant) to conduct comparative population genomics. They report 15 different mutations in the insecticide target site from resistant populations, many reported here for the first time, and look at the mutation and selection processes underlying the evolution of resistance, through GWAS, haplotype mapping, and testing for loss of diversity indicating selective sweeps. None of the target site mutations found in resistant populations was found in pre-exposure populations, suggesting that the mutations may have arisen de novo rather than being present as standing variation, unless initially present at very low frequencies; a de novo origin is also supported by evidence of selective sweeps in some resistant populations. Furthermore, there is no significant evidence of migration of resistant genotypes between the sampled field populations, indicating multiple origins of common mutations. Overall, this indicates a very high mutation rate and a wide range of mutational pathways to resistance for this target site in this pest species. The series of population genomic analyses carried out here, in addition to the evolutionary processes that appear to underlie resistance development in this case, could have implications for the study of resistance evolution more widely.Strengths:This paper combines phenotypic characterisation with extensive comparative population genomics, made possible by the availability of multiple population samples (each with hundreds of individuals) collected before as well as after the introduction of the pesticide cyatpyrafen, as well as lab-evolved lines. This results in findings of mutation and selection processes that can be related back to the pesticide resistance trait of concern. Large numbers of mites were tested phenotypically to show the levels of resistance present, and the authors also made near-isogenic lines to confirm the phenotypic effects of key mutations. The population genomic analyses consider a range of alternative hypotheses, including mutations arising by de novo mutation or selection from standing genetic variation, and mutations in different populations arising independently or arriving by migration. The claim that mutations most likley arose by multiple repeated de novo mutations is therefore supported by multiple lines of evidence: the direct evidence of none of the mutations being found in over 2000 individuals from naive populations, and the indirect evidence from population genomics showing evidence of selective sweeps but not of significant migration between the sampled populations.Weaknesses:As acknowledged within the discussion, whilst evidence supports a de novo origin of the resistance-associated mutations, this cannot be proven definitively as mutations may have been present at a very low frequency and therefore not found within the tested pesticide-naive population samples.

We agree that we could not definitively exclude the presence of a very low incidence of favoured mutations before the introduction of this novel acaricide.

Near-isofemale lines were made to confirm the resistance levels associated with five of the 15 mutations, but otherwise, the genotype-phenotype associations are correlative, as confirmation by functional genetics was beyond the scope of this study.

We hope that future functional studies will validate the effects of these mutations on resistance in both the two-spotted spider mite *T. urticae* and other spider mite species. This could be done by creating near-isogenic female lines or using CRISPR-Cas9 technology, as gene knockouts have recently been established for *T. urticae*.

**Recommendations for the authors:**

**Reviewer #1 (Recommendations for the authors):**
(1) Could the authors elaborate on the environmental context (e.g., climate, geography) of the sampled populations to give more nuance to the analysis of genetic differentiation and resistance evolution?

We have explored the influence of geographic isolation on the frequency of resistance alleles by Mantel tests (isolation by distance). We didn’t investigate the influence of climate, because most of the samples were from greenhouses, where the climate to which the pest is exposed is unclear.

(2) Line 161: is this supposed to be one R and one S?

Yes, we added this information (LabR and LabS).

(3) Line 207: variation is not saturated at the first two sites because the different combinations are not seen. This is a bit misleading.

What we wanted to indicate was that the two codon positions are saturated, rather than their combinations. We revised this sentence by adding “of each codon position”.

(4) Line 376: continuous selection did not "result in a new mutation arising". Rather, the mutation arose and was subsequently selected on.

We revised the expression of this de novo mutation and selection process.

(5) Line 402: can the authors explore what Ne would be necessary to drive the number of mutational origins they observe, as in (Karasov et al. 2010)?

It is challenged to estimate Ne, especially when mutation rate data from the two-spotted spider mite *T. urticae* is unavailable. We observed 2.7 resistant mutations per population in samples collected in 2024, seven years after the release of cyetpyrafen. The estimated mutation rate (Θ) is 0.0193, given 20 generations per year for *T. urticae*. An effective population size (Ne) of 2.29*10^6^ would be necessary to reach the number of de novo mutations observed in this study, given Θ = 3Neμ (haplodiploid sex determination of *T. urticae*) and a mutation rate of μ = 2.8*10^-9^ per base pair per generation as estimated for *Drosophila melanogaster* (Keightley et al., 2014). The high reproductive capacity of *T. urticae* (> 100 eggs per female) and short generation time makes it easier to reach such a population size in the field as we now note.

(6) Line 482: how did the authors precisely kill 60% of samples with their selection? What was the applied rate? In general, listing the rates of insecticide used in dose response would be useful to decipher if LD50s are projected outside of the doses used (seems like they are). In this case, authors should limit their estimates to those > the highest rate used in the dose response.

It is difficult to control mortality precisely. We applied cyetpyrafen every two generations but did not determine the LC_50_ every two generations. When mortality was lower than 60%, another round of spraying was applied by increasing the dosage of the pesticide. The LC_50_ values were tested at F_1_, F_32_, F_54_, F_60_, F_62_, and F_66_ generations to establish the trajectories around resistance.

(7) The light pink genomic region in Figure 2 was distracting. Why is it included if there is no discussion of genomic regions outside the sdh genes? Generally, there was a lot going on in this figure, and some guiding categories (i.e., lab selected vs wild population) on the figure itself could help orient the reader.

We included chromosome 2 colored in light pink/ red to show the selection signal across a wider genomic region. In the figure legend, we added a description of the lab selected, field resistant and field susceptible populations. Very little common selection signal was detected among resistant populations on chromosome 2, indicating this region was less likely to be involved in resistance evolution of *T. urticae* to cyetpyrafen. We also described the result briefly in the figure legend.

**Reviewer #2 (Recommendations for the authors):**
(1) The most significant aspect of this study is the use of multiple pest population samples taken before as well as after the introduction of a class of pesticides, allowing a thorough comparative population genomics study in a species where a range of resistance mutations have appeared within a few years. I would prefer to see a title conveying this significance, rather than the current study, which focuses on the total number of mutations and claimed notoriety of the (at that point unnamed) study species. Similarly, I would prefer an abstract that relies less on superlative claims and includes more details: the scientific name of the study species; the number of years in which resistance evolved; the number of historical specimens; how the resistance levels for single mutations were shown.

(1) The title was changed by adding “the two-spotted spider mite *Tetranychus urticae*” and removing the “unprecedented number” to emphasize that “recurrent mutations drive rapid evolution”, i.e., “Recurrent Mutations Drive the Rapid Evolution of Pesticide Resistance in the Two-spotted Spider Mite *Tetranychus urticae*.”

(2) The scientific name of the study species was added.

(3) The number of years in which resistance evolved was added.

(4) The number of historical specimens was added (2,317).

(5) Because we used homozygous lines but not iso-genic lines or gene-edited lines, our bioassay data could not provide direct evidence on the level of resistance conferred by each mutation. We revised our description of the results and removed this content from the abstract.

Line 29: if you want to claim the number is unprecedented, please specify the context: unprecedented for a pesticide target in an arthropod pest? (more resistance mutations may have been found in bacteria/fungi...).

We revised the sentence by adding “in an arthropod pest”.

Line 30: rather than a claim of notoriety, it may be better to specify what damage this pest causes.

Revised by describing it as an arthropod pest.

Line 34: please clarify, was this all in different haplotypes, or were some mutations found in combination?

Done: We identified 15 target mutations, including six mutations on five amino acid residues of subunit sdhB, and nine mutations on three amino acid residues of subunit sdhD, with as many as five substitutions on one residue.

(2) The introduction begins by framing the context as resistance evolution in invertebrate pests. However, the evolutionary processes examined in the study are applicable to resistance in other systems, and potentially to other cases of rapid contemporary evolution. The authors could show wider significance for their work beyond the subfield of invertebrate pests by including more of this wider context in their introduction and discussion: even if this means they can no longer claim novelty based on the number of mutations alone, the study is a strong example of the use of population genomics combined with functional and phenotypic characterisation to investigate the evolutionary processes underlying the emergence of resistance, so could have wider importance than within its current framing.

The background was revised as mentioned above to take this into account.

For example, in lines 48-50, please clarify what is meant by pesticides here (insects/arthropods? weeds and pathogens too?) In lines 69-73, the opposite is sometimes seen in fungal pathogens, with large numbers of mutations generated in lab-evolved strains.

We extended pesticides to those targeting arthropods, weeds and pathogens. We still emphasize the situation mainly with respect to arthropod pests.

(3) Lines 91-93: how many modes of action? How recently were SDHI acaricides introduced?

Added: at least 11 groups of acaricides based on their modes of action. SDHI was launched in 2007.

(4) Line 98-102: Use in China is a useful background for the study populations, but the global context should be included too.

Yes, four SDHI acaricides developed around the globe were introduced.

(5) Line 113: They show diverse mutations, but all within the mechanism of target-site point mutations.

We agree to your suggestion. This sentence has been removed as it repeats information stated above it.

(6) Line 115-116: Yes, agreed; I think this is the main strength of the current study and should be emphasised sooner.

Thanks.

(7) Line 158: Selective sweep signals were clear in half of the resistant populations but not in the others. The suggestion that the others had undergine soft sweeps, with multiple mutations increasing in frequency simultaneously but no one reaching fixation, seems reasonable; but the authors could compare the populations that did show a sweep with those that did not (for example, was there greater diversity or evenness of genotypes in those that did not?).

Five resistant populations with selection signals identified by PBE analysis (Figure 2b) showed corresponding decreases in *π* and Tajima’s *D* near the two SDH genes but not across the genome (Figure S1).

(8) Line 313: please clarify "in combination with other mutations" within a mixed population or combined in one individual/haplotype? Also, the phrase "characterised the function" may be a little misleading, as this is a correlative analysis, not functional confirmation.

None of the combinations of different resistant mutations was observed in a single haplotype. Here, we examine resistance levels associated with a single mutation or two mutations on *sdhB* and *sdhD* in one individual, i.e. sdhB_I260V and sdhD_R119C. We revised the sentences to avoid any implication of functional confirmation.

(9) Line 358: again, please clarify the context: among arthropod pests?

Done.

(10) Line 360-363: please give some background on when and where these related compounds were introduced.

Added.

(11) Line 410: yes fitness costs may be a factor, but you could also give an example of a cost expressed in the absence of any pesticides, as well as the given example of negative cross-resistance.

We added the example of the H258Y mutation which causes both fitness costs and negative cross-resistance.

(12) Lines 419-438: this is one aspect where the situation for insecticides is in contrast with some other resistance areas.

Yes, we restricted these statements to arthropod pests.

(13) Line 466: some more detail could be given here: for example, SNP-specific monitoring would be less effective, but amplicon sequencing would be more suitable.

Yes, revised.

(14) Lines 472-475: Please list the numbers of field/lab, pre/post exposure, and sensitive/resistant populations within the main text.

Done. The number of sensitive/resistant populations was reported in the result section.

(15) Line 483: randomly selected individuals?

Yes, added randomly selected individuals.

(16) Line 556: Sanger sequencing to characterise populations? Or a number of individuals from each population?

Revised.

(17) References: there are some duplicate entries, please check this.

Checked.

(18) Figure 1e: consider a log(10) scale to better show large fold changes and avoid multiple axis breaks.

Thanks for your suggestions. However we didn’t scale the LC_50_ value, because we wanted to show the specific impact of 1,000 mg/L. The breaks in the Y axis around 30 mg/L -1,000 mg/L reveal that the LC50s of the resistant populations were all greater than 1000 mg/L, while those of the susceptible populations were all below 30 mg/L. This justified the use 1000 mg/L as a discriminating dose to investigate resistance status and level in subsequent work.